# East Antarctic warming forced by ice loss during the Last Interglacial

**David K. Hutchinson** [1,2] ✉, **Laurie Menviel** [1,2], **Katrin J. Meissner** [1,3] & **Andrew McC. Hogg** [3,4]

During the Last Interglacial (LIG; 129-116 thousand years before present), the Antarctic ice sheet (AIS) was 1 to 7 m sea level equivalent smaller than at pre-industrial. Here, we assess the climatic impact of partial AIS melting at the LIG by forcing a coupled climate model with a smaller AIS and the equivalent meltwater input around the Antarctic coast. We find that changes in surface elevation induce surface warming over East Antarctica of 2 to 4 °C, and sea surface temperature (SST) increases in the Weddell and Ross Seas by up to 2 °C. Meltwater forcing causes a high latitude SST decrease and a subsurface (100–500 m) ocean temperature increase by up to 2 °C in the Ross Sea. Our results suggest that the combination of a smaller AIS and enhanced meltwater input leads to a larger sub-surface warming than meltwater alone and induces further Antarctic warming than each perturbation separately.

During the last interglacial (LIG), summer temperatures over land areas of the northern high-latitudes were 4 to 5 °C higher than the pre-industrial (PI), primarily due to differences in orbital forcing[1–3]. In contrast, greenhouse gas forcing was reduced, with $CO_2$, $CH_4$ and $NO_2$ concentrations all estimated to be slightly lower than during PI[3,4]. Antarctica's contribution to the LIG sea level high-stand is estimated to have occurred between 129.5 and 124.5 ka, with a peak at 127 ka, while Greenland's contribution slowly ramped up from 127 ka onwards[5]. The Greenland ice sheet is estimated to have contributed 0.4 to 4.4 m sea-level equivalent (SLE)[6,7] compared with 1 to 7 m SLE for Antarctica[5,8,9]. A recent study suggests the AIS contributed 5.7 m SLE of sea level rise, with a likely range of 3.6 to 8.7 m (68% probability)[10]. However, the evolution of these ice sheets and their associated climate feedbacks are not well understood, motivating further examination with physical climate models.

Paleo-proxy records suggest 1.8 ± 0.8 °C higher Southern Ocean summer sea surface temperatures (SSTs), and annual mean SSTs up to 3 °C higher at 127 ka compared with PI at ~40°S[11,12]. Recent coordinated efforts to model the LIG at 127 ka, under the Paleoclimate Model Intercomparison Project 4 (PMIP4) underestimate the warm anomaly suggested by Southern Hemisphere SST proxy records[13]. Most of the LIG PMIP4 experiments suggest an anomaly of 1 to 2 °C in austral winter south of 60°S, while simulating no significant changes in austral summer, thus leading to an annual mean SST change of −0.5 to +2 °C[13]. The LIG 127 ka experiments also underestimate the annual mean surface air temperatures over Antarctica. Ice core records suggest an anomaly of ~2.2 °C compared with PI[11,14], however the multi-model mean finds an increase of ~0.5 °C[13]. One of the reasons for these discrepancies might be the absence of changes in the Antarctic ice sheet in the PMIP4 experiments, which are expected to play an integral role in high latitude temperature change. Coupling a dynamic Antarctic ice sheet with the ocean-atmosphere-sea ice-land components necessary for PMIP4 simulations remains a significant technical barrier, primarily due to the problem of ice-ocean coupling necessary for a large marine-based ice sheet[15]. Recent progress has been made towards coupling a dynamic ice sheet with a fully coupled climate model[16], however this approach is still under development, and remains challenging to adapt to a paleoclimate scenario such as the LIG. Therefore, to investigate the effects of changing ice sheets, we currently prescribe ice sheet changes through the model boundary conditions.

Ice-sheet modelling studies suggest differing mechanisms that could account for the observed LIG sea level rise, but all suggest that a

[1]Climate Change Research Centre, University of New South Wales, Sydney, NSW, Australia. [2]The Australian Centre for Excellence in Antarctic Science, University of New South Wales, Sydney, NSW, Australia. [3]ARC Centre of Excellence for Climate Extremes, University of New South Wales, Sydney, NSW, Australia. [4]Research School of Earth Sciences, Australian National University, Canberra, ACT, Australia. ✉e-mail: david.hutchinson@unsw.edu.au

sub-surface warming of the Southern Ocean is necessary for AIS retreat, with a potential collapse of the West Antarctic ice sheet (WAIS) above 2–3° subsurface warming[17]. A recent model study suggests that subsurface warming induced by weakening of the Atlantic meridional overturning circulation (AMOC) can cause an Antarctic ice mass loss of 3.42 m SLE by the early LIG[18]. At the higher end of ice loss estimates, an Antarctic ice-sheet modelling study was able to simulate a 7.5 m SLE increase from a disintegration of the Antarctic ice sheet (AIS) at the LIG, by forcing their model with a 3 °C sub-surface warming and implementing a marine ice-cliff instability (MICI) parametrisation[19]. However, constraints on MICI remain uncertain, since previous sea level rise events can be explained within uncertainty bounds without invoking MICI[20]. Furthermore, in the recent Ice Sheet Model Intercomparison (ISMIP6) projections of 21st century Antarctic ice sheet evolution, none of the modelling groups included a MICI parameterisation[21].

A limitation of current LIG studies is that either the climate is studied without including potential Antarctic ice-sheet (AIS) changes or the AIS evolution is assessed using climate model outputs as a prescribed (non-interactive) boundary condition. For example, a previous coupled climate model study assessed the processes leading to the LIG $\delta^{18}O$ maximum and found that this could not be explained by WAIS retreat, but instead could reflect sea ice loss[22]. A related approach is to use iterative coupling (i.e. repeated updates of boundary conditions) between an ice sheet model and a climate model; such a method has found significant positive feedbacks between subsurface warming and Antarctic ice melt for future warming scenarios[23], but has not been implemented for the LIG. Meltwater input into the Southern Ocean at the LIG has been shown to lead to an increase in Southern Ocean stratification and thus a subsurface warming[24]. This positive feedback could enhance Antarctic ice-sheet disintegration[25,26].

The impact of the WAIS removal on $\delta^{18}O$ was previously studied in a coarse-resolution (2.5° × 3.75° ocean and atmosphere) climate model[22], assuming an idealised flattening or removal of the WAIS. In that study the oceanic response was not studied in detail. To fill this gap, here we assess the response to a more widely distributed ice sheet perturbation and meltwater forcing informed by ice sheet modelling, rather than just WAIS removal, and we study more closely the ocean circulation and climate responses.

Here we assess the impact of a partial removal of the AIS upon Antarctic and Southern Ocean climate. This partial removal of ice consists of separate and combined contributions from changing the ice sheet topography and from enhanced meltwater forcing. Our aim is to investigate how feedbacks from partially melting the Antarctic ice-sheet would impact Southern Ocean and Antarctic climate, especially in regions that affect the stability of the remaining Antarctic ice sheet.

## Results
### Climate response to a partial AIS removal
We investigate the impact of a partial removal of the AIS by performing an experiment under LIG boundary conditions in which the AIS elevation and extent, as well as ocean bathymetry were modified to represent a 4.1 m and 7.1 m SLE Antarctic ice mass loss[19,27] (Fig. 1 and Supplementary Fig. S1, Methods; experiments SL4.1 and SL7.1). To highlight the distribution of ice loss around Antarctica, we divide the SLE perturbation into 60° sectors of longitude (shown in Fig. 1a, b). We also calculate the contribution from the WAIS, the East Antarctic Ice Sheet (EAIS), and the Antarctic Peninsula, as defined by drainage basin definitions of Antarctica[28]. These amounts are shown in Supplementary Dataset 3. The key difference between the two scenarios is found in the EAIS, where the ice loss is 1.92 m SLE in SL4.1, increasing to 4.63 m SLE in SL7.1.

In both the SL4.1 and SL7.1 cases, partial ice sheet removal leads to a 0.5–1.5 °C annual mean SST increase in the Weddell Sea, extending towards the Indian sector of the Antarctic shelf (30°W–30°E), as well as

in the Ross Sea (180–150°W; Supplementary Fig. S2a, d). Both the Weddell and Ross Sea warming occur predominantly in summer, in combination with a retreat in summer sea ice (Supplementary Figs. S3 and S4), though an expansion of sea ice occurs in the Weddell Sea in winter. Those areas of sea ice expansion are associated with a slight decrease in annual mean temperature (between −0.25 and 0 °C). East Antarctic annual mean coastal SSTs increase by around 0.5–1 °C from 0° to 60°E in combination with sea ice retreat, with a stronger signal in the SL7.1 case than the SL4.1 case. However, in other regions the SL4.1 and SL7.1 simulations show minimal differences and even some stronger anomalies in SL4.1. From 60 to 180°E, SST anomalies are close to zero near the coast, with the sea ice edge mostly unchanged (Fig. 2a, d). The summer warming is driven by ice-albedo feedbacks both from replacing part of the marine ice sheet with ocean, and from a retreat in summer sea ice in those sectors.

Some of the SST anomalies extend to the subsurface (Supplementary Fig. S2d), with a 0.25–0.5 °C subsurface warming at 100–500 m depth simulated along the East Antarctic shelf between 0 and 180°E in both SL4.1 and SL7.1. A -1 °C SST cooling is simulated at 60 to 65°S in the Amundsen and Bellingshausen Seas in both SL4.1 and SL7.1 (Fig. 2a, d). This signal is also mirrored in a 1–2 °C subsurface cooling in these regions, with a slightly stronger cooling signal in SL4.1 than SL7.1 (Fig. S2a, d). This regional cooling is associated with a northward shift of the subpolar front of the Antarctic Circumpolar Current (ACC), which steers colder isotherms to the north (Figure S5). Changes in the pathways of the ACC also manifest in subsurface warming of >2 °C around 140–180°W, 45–50°S (Supplementary Fig. S5), as well as subsurface warming around -1 °C located at 45°S in the Indian Ocean sector. The shifts in the ACC are in turn driven by both changes to the Antarctic coastal bathymetry when the ice is removed, and changes to the wind field, notably a weakening of the westerly winds in all of the Pacific sector as well as some of the Indian sector of the Southern Ocean, with a slightly larger anomaly in SL4.1 than SL7.1 (Supplementary Fig. S6). This weakening is likely due to high-latitude surface warming causing a decrease in baroclinic instability, which in turn decreases the strength of the midlatitude westerly winds[29]. In addition, mixed layers deepen in the Ross Sea from less than 50 m in the LIG control to 400–500 m in the SL4.1 experiment, and to some extent in the SL7.1 experiment (Supplementary Fig. S7). This deepening is driven by greater exposure of higher latitude shelf regions to atmospheric cooling, enabling denser water to form in the Ross Sea region. Sea-ice retreat in the Atlantic-Indian sector also leads to greater exposure of surface waters to winter cooling, thus deepening the mixed layers between 0 and 30°E (Supplementary Fig. S7). This surface forcing increases Antarctic Bottom Water (AABW) formation from 9.8 Sv in the LIG control to 13.0 Sv in SL4.1 and 11.2 Sv in SL7.1 (Figs. 3 and 4).

An important feature of our simulations is the ability to model the Antarctic continental climate response to a partial loss of the ice sheet. There are four proxy records from -127 ka that record Antarctic surface air temperature (SAT) anomalies of 0.9 to 3.3 °C with respect to PI[11,30]. These Antarctic temperature anomalies are not captured by recent PMIP4 experiments that simulate the LIG climate without explicitly including Antarctic ice loss. Our LIG control experiment produces a similar result to the PMIP4 ensemble; Antarctic SAT does not change significantly compared with PI (Supplementary Fig. S9e). In contrast, the ice loss experiment SL7.1 captures an annual mean SAT increase of 2 to 4 °C in the Antarctic continental interior, in good agreement with proxy data (Fig. 5b). The SL4.1 experiment also shows East Antarctic warming in the continental interior, although to a lesser extent than SL7.1. These changes are closely linked to reductions in ice elevation in East Antarctica (Fig. 1a, b), indicating lapse rate effects have a major impact in explaining this warming signal. However, some of the warming signal is also due to ice-albedo effects warming the newly formed ocean surface (i.e. loss of marine ice sheet), allowing

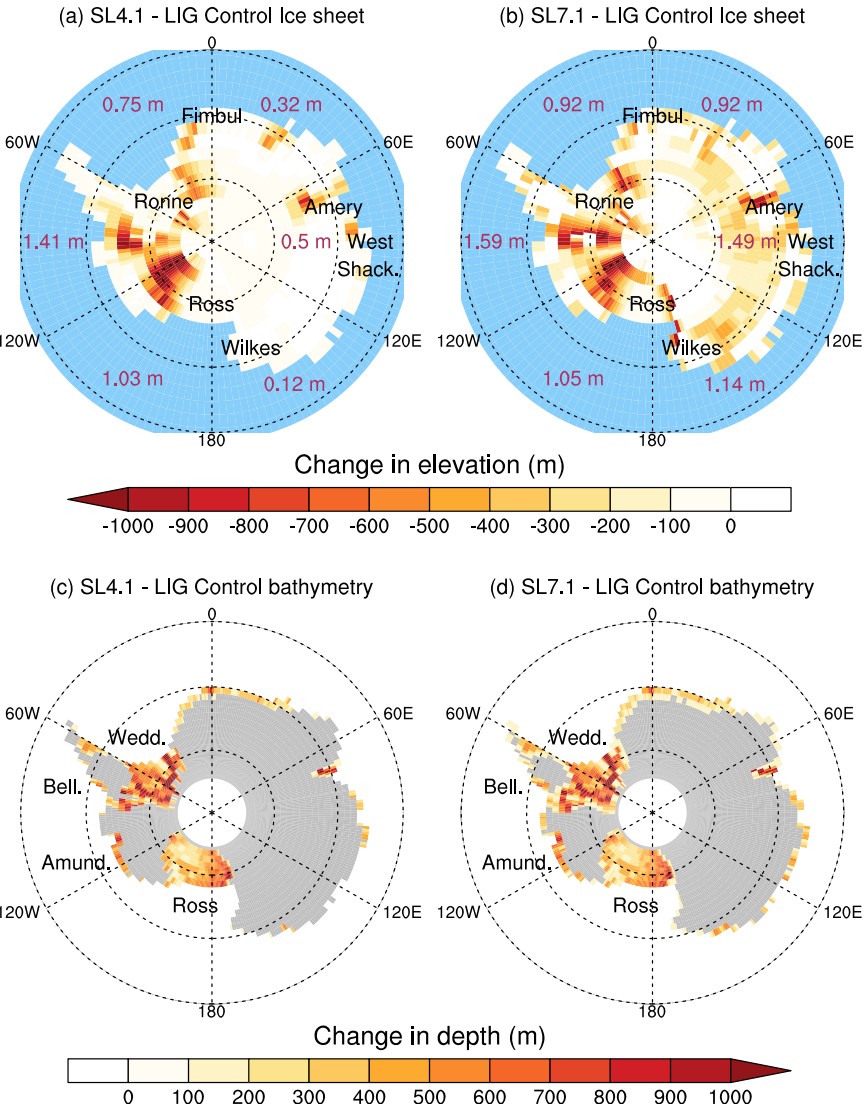

**Fig. 1 | Changes to topography and bathymetry in the perturbation experiments.** Anomalies of land-based topography in the (**a**) SL4.1 configuration and (**b**) SL7.1 configurations; anomalies of bathymetry in the (**c**) SL4.1 and (**d**) SL7.1 configurations. The maroon numbers in **a** and **b** indicate the ice loss in m SLE for each 60° sector of longitude (separated by dashed meridians). Locations of prominent ice shelves or ice sheets are labelled in **a** and **b**, while Antarctic Seas mentioned in this study are labelled in **c** and **d**. Abbreviations: Shack Shackleton, Amund Amundsen, Bell Bellingshausen, Wedd Weddell.

additional heat to be transported into the continent via the atmospheric circulation.

Surface wind anomalies indicate a decrease in the predominant (southerly) offshore winds off East Antarctica in both the SL4.1 and SL7.1 experiments (Supplementary Fig. S6b, e). This decrease in southerly winds enables warmer oceanic airmasses to reach further inland over East Antarctica. Surface wind anomalies also indicate a minor enhancement of easterly winds along the Antarctic coast (Supplementary Fig. S6). We suggest that these wind patterns may be driven by two factors. First, the weakening of the equator-to-pole gradient (due to Antarctic surface warming) drives a weakening of the midlatitude westerlies, since polar warming at the surface has been found to reduce baroclinic instability and therefore weaken the midlatitude westerlies[29]. Second, the lowering of topographic barriers especially in West Antarctica allows a greater exchange of surface winds into near-polar latitudes. In reality, these topographic changes would also impact the katabatic winds, but these remain poorly resolved with an atmosphere resolution of ~2°. The southward shift of the easterlies also strengthens the Antarctic Slope Current (ASC;

Supplementary Fig. S8). The new ocean gateway at 73°S, 86–80°W opened by the removal of ice sheet, allows a southward transport of 0.7 Sv and 1.1 Sv respectively in SL4.1 and SL7.1 through the gateway from the Pacific to the Atlantic sector.

## Climate response to meltwater input around Antarctica

We further investigate the impact of meltwater forcing around Antarctica, equivalent to 4.1 m and 7.1 m SLE of AIS disintegration (Methods, experiments FW4.1 and FW7.1). This meltwater input leads to approximately 1–2 °C SST decrease from 150 to 30°W off the coast of Antarctica in the FW4.1 case (Fig. 2b). A similar pattern, but stronger magnitude of SST decrease (2–3 °C) occurs in the FW7.1 case. This cooling is associated with a large expansion of sea ice (Fig. 2). These changes are driven by the freshwater forcing creating a stronger halocline near the surface, which ensures that surface waters remain stably stratified under cooling conditions. A more stable halocline enables both greater sea ice formation, and a reduction of deep ocean convection in the Weddell Sea as indicated by changes to mixed layer depths (Supplementary Fig. S7). The rate of Southern Ocean deep

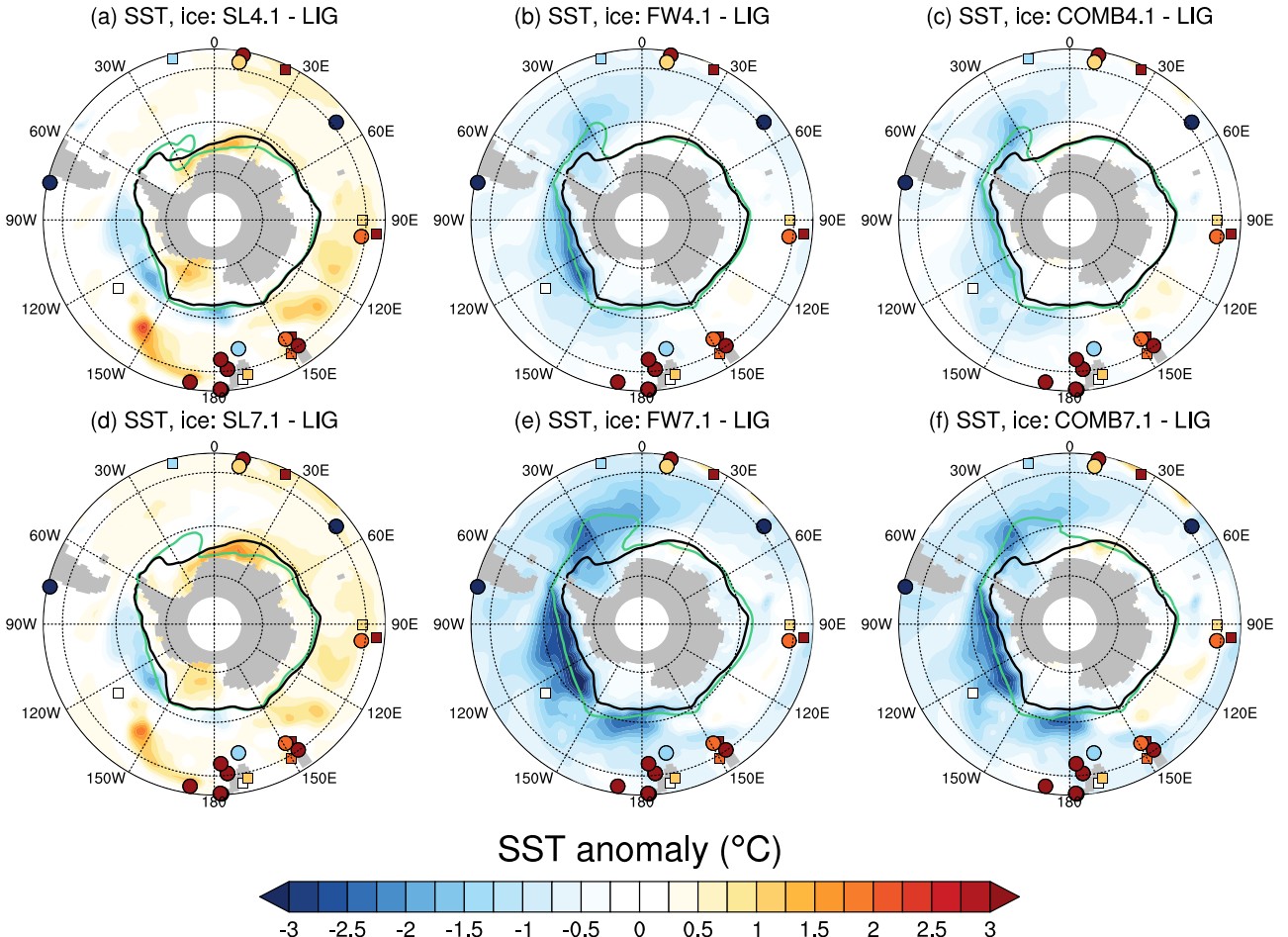

**Fig. 2 | Annual mean sea surface temperature (SST) anomalies in the perturbation experiments, and SST proxy data.** SST anomalies for the (**a**) SL4.1, (**b**) FW4.1, (**c**) COMB4.1 experiments; and (**d**) SL7.1, (**e**) FW7.1 and (**f**) COMB7.1 experiments compared with the LIG control. Contours show the annual mean sea ice edge, denoting 15% concentration for the LIG (black) and the perturbation experiment (green). The coloured circles represent proxy SST data from refs. 11,12 as used in PMIP4, while the coloured squares represent SST anomalies from 126 ka from ref. 32.

water reduces from 9.8 Sv in the LIG control, to 7.8 Sv in the FW4.1 case and 5.8 Sv in the FW7.1 case (Figs. 3 and 4).

Due to the increased surface stratification and reduced deep ocean convection, subsurface (100–500 m depth) temperatures in the FW4.1 experiment increase by up to 2 °C in the Ross Sea (Supplementary Fig. S2b), with a slightly stronger increase in FW7.1 (Supplementary Fig. S2e). The meltwater-induced stratification isolates the surface from the warmer circumpolar deep water (CDW), which is then transported further south and warms Antarctic subsurface waters. The increased southward advection of CDW is indicated by a flattening of neutral density surfaces in the high southern latitudes, enabling southward isoneutral transport of warmer deep waters (Figs. 3 and 4). This effect is similar to that found in previous coupled climate model simulations of the LIG[24] and future warming scenarios[31], which found that meltwater perturbations tended to lower Antarctic SST while warming the subsurface.

Meltwater forcing also reduces Antarctic SAT, similar to the SST response (Fig. 5b, e). This perturbation in itself does not produce a good agreement with paleo-proxy records from East Antarctica, which show a warm anomaly compared with PI[11,30]. The strength of mid-latitude westerly winds increases in the FW4.1 and FW7.1 experiments (Supplementary Fig. S6c, f), likely due to an enhanced equator-to-pole gradient. This is the opposite response of the same mechanism seen in the SL4.1 and SL7.1 wind changes, in which surface polar warming reduces baroclinic instability[29].

## Climatic response to a combined partial AIS removal and enhanced meltwater input around Antarctica

Finally, we assess the climatic impact of the combined ice sheet removal and meltwater inputs of 4.1 m and 7.1 m SLE (Methods, experiments COMB4.1 and COMB7.1). These combined experiments lead to mostly warming over Antarctica and cooling over the Southern Ocean, resembling a combination of SAT anomalies from both the ice loss and meltwater experiments (Fig. 5c, f). While the COMB7.1 experiment shows some modest warming in the four proxy locations[11,12,30], the influence of lower SSTs due to the meltwater forcing produces a poorer agreement with the proxy data in the combined cases than in the ice loss-only cases. A quantitative comparison between the proxy data, and the modelled SAT changes at each proxy location is shown in Supplementary Dataset 1.

The meltwater forcing dominates the ocean's surface response with an increase in surface ocean stratification all around Antarctica, and a decrease in deep-water formation in the Weddell Sea (Supplementary Fig. S7). The annual mean sea-ice extent is 4.7% and 12.8% larger than the LIG control in COMB4.1 and COMB7.1 respectively (here we count grid cells only if they exist as ocean in both simulations). The sea ice expansion is associated with 1–1.5 °C and 1.5–2 °C SST decreases in the Weddell Sea in COMB4.1 and COMB7.1 respectively (Fig. 2c, f). Similarly, there is a -1 °C and -2–3 °C decrease in the Amundsen and Bellingshausen Seas in COMB4.1 and COMB7.1 respectively. Along the East Antarctic coast, SST change is slightly negative ( ~ -0.5–0 °C),

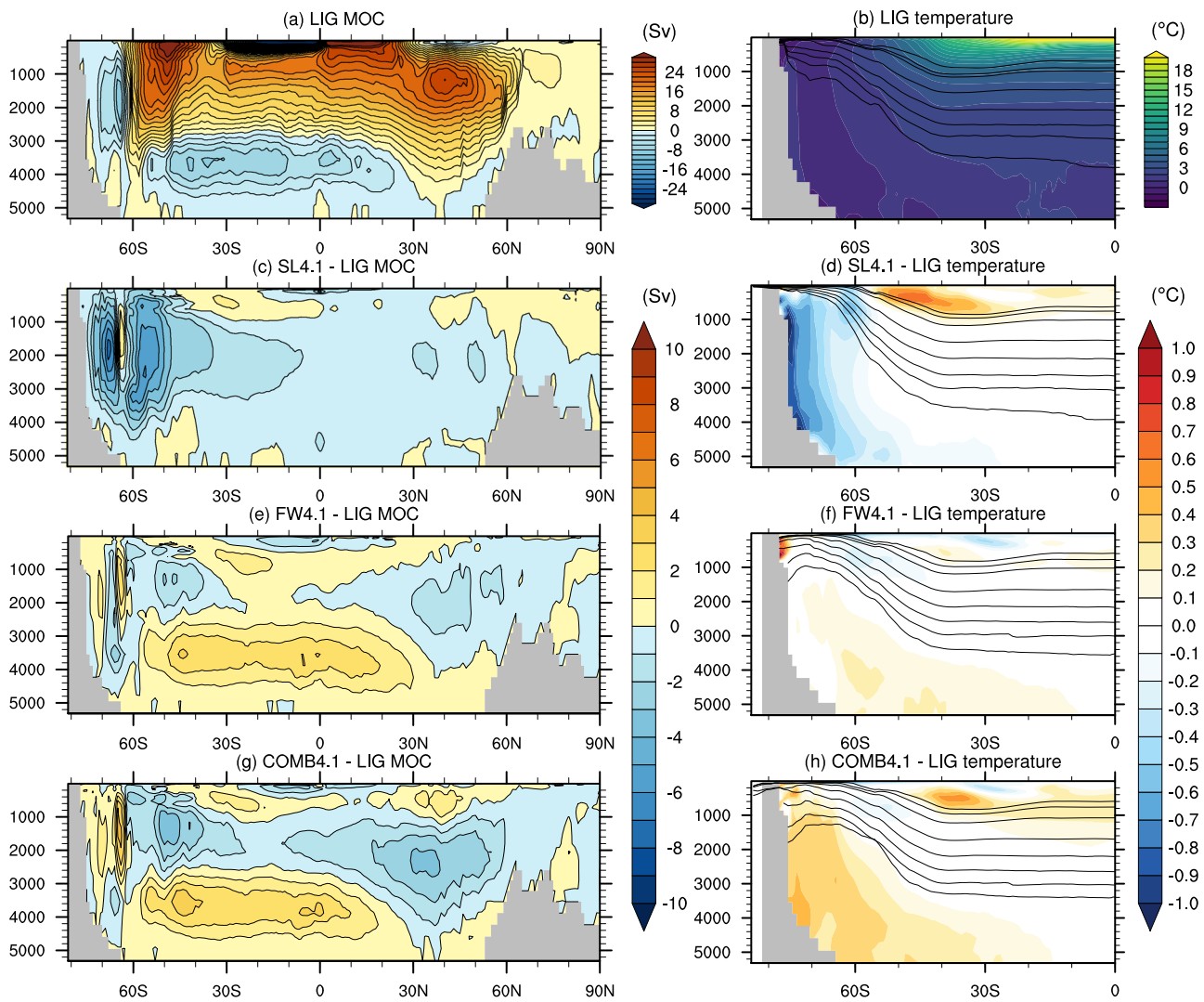

**Fig. 3 | Meridional overturning circulation (MOC) and zonal mean temperatures for the 4.1 experiments. a, b** Absolute values in the LIG experiment; and anomalies in the (**c, d**) FW4.1, (**e, f**) SL4.1 and (**g, h**) COMB4.1 experiments. Contours on the right-hand plots are ω-surfaces of neutral density, along which strong mixing occurs (see Methods[48]).

though it is warmer than the meltwater simulations in regions where ice sheet is converted to ocean (Fig. 1). A comparison of annual mean SST proxy data[11,12,32] from the Southern Ocean with model anomalies is given in Supplementary Dataset 1, and plotted in Fig. 2, while summer (DJF) average SST proxy data[11] and model anomalies are compared in Supplementary Dataset 2. These comparisons indicate that the simulated SSTs are generally too low compared to the SST proxy data. Most of the Southern Ocean SST proxies indicate warming of at least several degrees, although a few sites also indicate strong cooling (Fig. 2), with the proxy anomaly data ranging from −6.8 °C to 11.5 °C (Supplementary Dataset 1). In general, the best agreement is found in the ice loss only experiments (SL4.1 and SL7.1), while a poorer agreement is found when introducing meltwater both separately (FW4.1 and FW7.1) and combined with the ice loss (COMB4.1 and COMB7.1). These data suggest that, in the context of uncertain timing of Antarctic melting at the LIG[5,10], that the peak meltwater pulse in the Southern Ocean did not occur simultaneously with the warmest SST anomalies, since meltwater tends to strongly cool the surface of the Southern Ocean.

The subsurface response in COMB4.1 also strongly resembles the pattern of the FW4.1 experiment. A 2 °C subsurface warming occurs in the Ross Sea. Elsewhere around the Antarctic coast, there is modest warming (<0.5 °C), but the COMB4.1 experiment shows some slight

additional warming compared to the meltwater-only case, similar to SL4.1. Compared to SL7.1, in COMB7.1 the Ross Sea also exhibits strong subsurface warming (>2 °C), but there is far more widespread subsurface warming around all of East Antarctica (0.5−1 °C). With the larger (7.1 m SLE) perturbations, the ice removal and meltwater effects compound each other more strongly.

Below 1000 m, the ice loss and meltwater effects compound nonlinearly. In the deep Southern Ocean, zonal mean temperatures show a clear cooling in SL4.1, and a modest warming in FW4.1 (Fig. 3d, f). However, the combined experiment COMB4.1 shows a stronger and larger warming signal than FW4.1, i.e. the two perturbations do not combine linearly (Fig. 3h). There is a large region of warming (0.2−0.4 °C) that extends from Antarctica to 50−60°S at intermediate depths, and up to 30°S below 4000 m (Fig. 3). Likewise, in COMB7.1, the deep Southern Ocean shows stronger warming (0.5−1.0 °C) and greater spatial extent than FW7.1, despite the fact that the ice loss experiment (SL7.1) cools the deep ocean; i.e. a non-linear warming effect.

AABW formation is reduced to 6.0 and 4.7 Sv in the COMB4.1 and COMB7.1 respectively, representing even stronger reductions than in the meltwater-only cases (7.8 and 5.8 Sv for FW4.1 and FW7.1). Deep water formation remains in the Atlantic sector, but shifts north-east

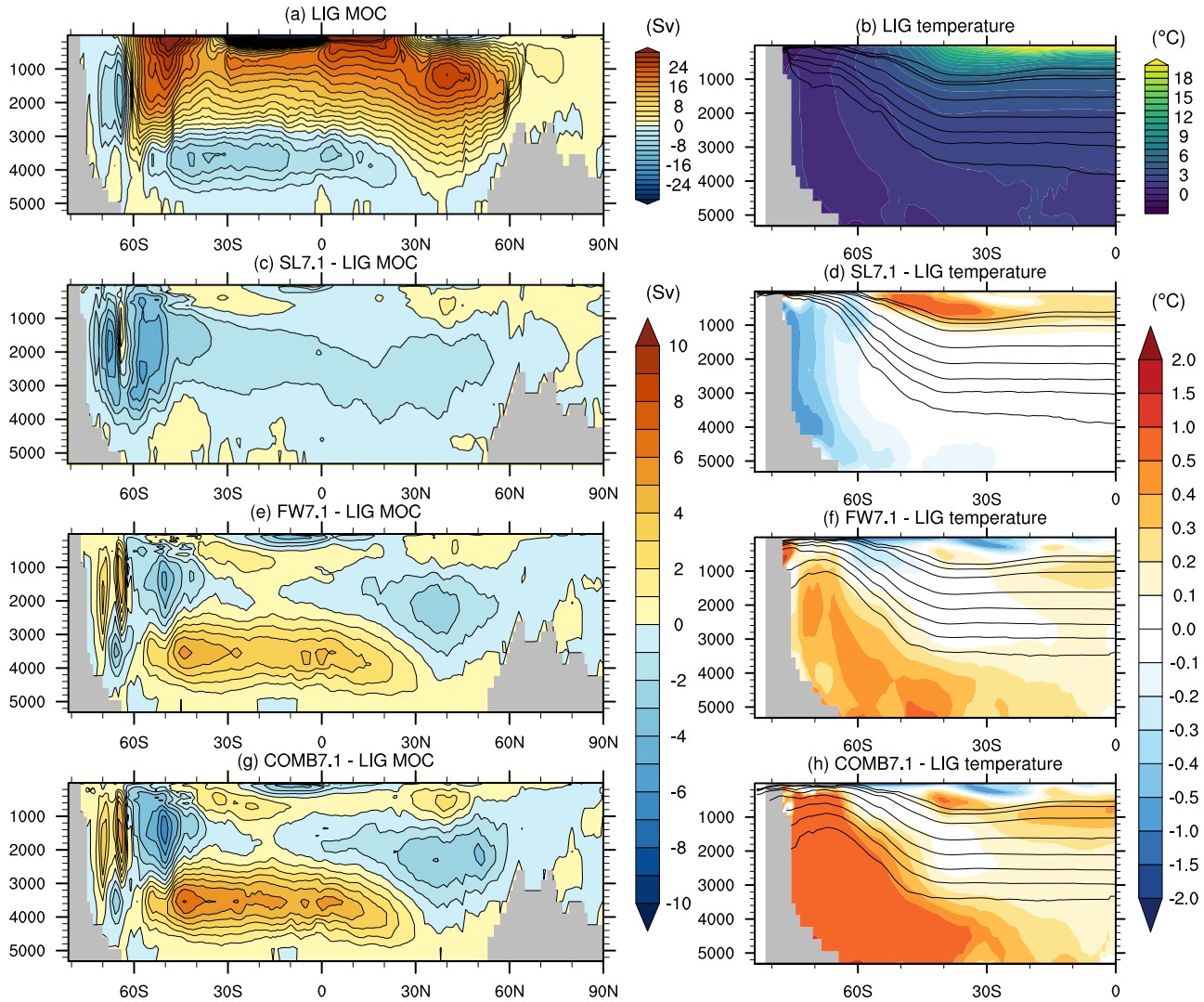

**Fig. 4 | Meridional overturning circulation (MOC) and zonal mean temperatures for the 7.1 experiments. a**, **b** Absolute values in the LIG experiment; and anomalies in the (**c**, **d**) FW7.1, (**e**, **f**) SL7.1 and (**g**, **h**) COMB7.1 experiments. Contours on the right-hand plots are ω-surfaces of neutral density, along which strong mixing occurs (see Methods[48]).

out of the Weddell Sea, and is absent from the Ross Sea. There is a clear flattening of isoneutral slopes in the combined perturbation experiments, as in the meltwater case, such that relatively warmer deep water is brought into the Southern Ocean. We find that the combination of flatter isoneutral slopes, a decrease in ventilation and SST warming from the ice loss perturbation produces this compound warming of the deep ocean. This is in contrast to the ice loss only cases (SL4.1 and SL7.1), where the steepening of isoneutral slopes means that the deep ocean is more strongly connected with (colder) high latitude atmosphere forcing.

The new ocean gateway at 73°S, 86–80°W opened by the removal of ice sheet allows a northward transport of 0.1 Sv and 1.0 Sv in COMB4.1 and COMB7.1 respectively through the gateway (i.e. Atlantic to Pacific sector; Supplementary Fig. S8d, g), which is in the opposite direction as the transport in SL4.1 and SL7.1. This throughflow does not greatly affect the surface temperature and salinity in the deep water formation regions.

## Discussion

This study has demonstrated the contrasting climatic impacts of Antarctic ice removal and Southern Ocean meltwater input. Previous climate model studies have shown that meltwater addition to the

Antarctic shelf induces an air temperature decrease over Antarctica[31] and a decrease of Southern Ocean SSTs[24,33,34]. Taken in isolation, this meltwater-induced cooling is at odds with LIG Antarctic air temperature proxies which suggest 0.9–3.3 °C higher temperatures compared to PI[30]. In contrast, our ice removal perturbation experiments display a 2–4 °C air temperature increase over East Antarctica, in good agreement with paleo-proxy records[30] (Supplementary Dataset 1). Our results contrast with a previous modelling study of the LIG, which found that a disintegration of the WAIS led to disagreement with δ[18]O records from East Antarctica[22]. As our model does not include oxygen isotopes, we cannot directly compare to the primary results of ref. 22, but we find that the positive temperature anomaly implied by the East Antarctic proxies could be explained by ice sheet reductions. Our results suggest that removing part of the AIS may lead to warming of East Antarctica by reducing topographic barriers, which reduces southerly offshore winds over East Antarctica.

Enhanced meltwater fluxes into the Southern Ocean dominate the ocean's response, and lead to an increase in the ocean's stratification which separates the colder, fresher surface waters from the warmer, saltier circumpolar deep waters below. As such, Southern Ocean SSTs decrease, particularly in the Weddell Sea. This cooling generally does not agree with Southern Ocean SST proxies from the LIG, which may

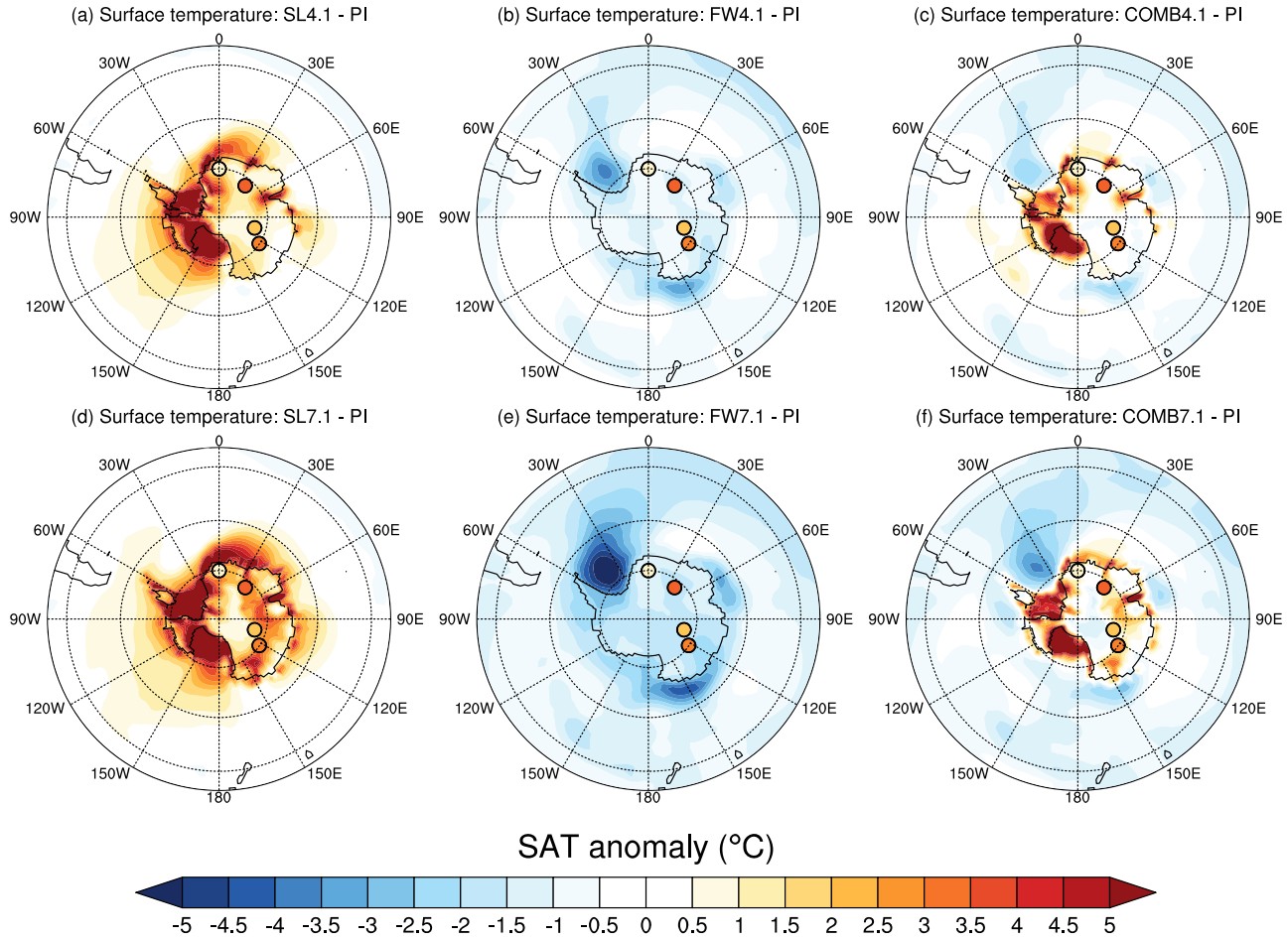

**Fig. 5 | Annual mean surface air temperature (SAT) anomalies in the perturbation experiments, and SAT proxy data.** SAT anomalies in the (**a**) SL4.1, (**b**) FW4.1, (**c**) COMB4.1 experiments; and (**d**) SL7.1, (**e**) FW7.1 and (**f**) COMB7.1 minus the PI experiment. The coloured circles represent proxy SAT anomalies from East Antarctica[11,30].

indicate that the peak in meltwater from Antarctic ice melt at the LIG possibly succeeded the peak positive SST anomalies in the Southern Ocean. We note, however, that Southern Ocean SST proxies are almost all located between 40 and 50 °S, while most of the SST anomalies simulated in our experiments are located poleward of 50 °S, where no proxy records are available.

Surface freshening leads to a subsurface warming in both the Ross Sea and around the East Antarctic coast of up to 2 °C, and 1 °C, respectively. In addition, the response of the deep ocean is strongly non-linear. While the meltwater perturbations cause warming of the deep Southern Ocean, and the ice loss perturbations cause cooling, the combined perturbations show a markedly stronger warming than either case, a counter-intuitive result that demonstrates the utility of forcing these perturbations both separately and together. We suggest this combined result is due to an enhanced stratification resulting from both surface freshening, and warming due to ice loss, as evidenced by deep water formation being reduced the most in the combined forcing cases.

The subsurface warming identified here can provide a positive feedback to Antarctic ice-sheet disintegration through marine ice sheet instability acting upon a reverse-sloping bedrock beneath the grounding line[35]. While the predominant regions of reverse-sloping bedrock are located in West Antarctica, there are also areas of East Antarctica where reverse-sloping bedrock occur, including the Amery (67–75°E), Shackleton (95–105°E) and Fimbul (3°W–3°E) ice shelves, and the Wilkes Subglacial Basin (140–150°E)[36], which may all be vulnerable to a marine ice-sheet instability feedback. In our SL4.1 and

SL7.1 simulations, there are strong localised SAT anomalies of upwards of 5 °C over the Amery and Fimbul ice shelf regions, due to significant melting of those shelves implied in the boundary conditions. Smaller SAT anomalies are found over the Shackleton ice shelf (1–2 °C) and the Wilkes Subglacial Basin (0.5–1.5 °C), due to modest ice reductions in those regions.

Some sedimentological and geochemical records suggest that the Wilkes Subglacial Basin, containing 3–4 m SLE of marine-based ice, may have retreated substantially during the LIG[37]. Furthermore, observations of authigenic uranium in the South Atlantic suggest an AABW decrease at around 127 ka[38], which could have been triggered by a meltwater pulse similar to the mechanism described here. However, maximal estimates of ice loss from the EAIS are constrained by the existence of continuous ice records[39], and [234]U accumulation[40] in and adjacent to the Wilkes Subglacial Basin. The lack of a large melting signal in these records suggests an upper bound of 0.8 m SLE ice loss for the Wilkes Subglacial Basin[39]. Our simulations are broadly consistent with this constraint: The ice perturbations in the 60° sector between 120 and 180°E (Fig. 1a, b), which covers a much larger area than the Wilkes Subglacial Basin, are 0.12 m and 1.14 m SLE in the SL4.1 and SL7.1 experiments respectively.

Previous ice sheet modelling studies of Antarctica have either produced a sea-level rise at the lower end of observations[17,18], or have invoked marine ice cliff instability[19], a mechanism which remains controversial. Our study shows that feedbacks from ice loss upon the coupled climate system may help resolve this discrepancy. The combined effects, especially on the subsurface temperature, suggest that a

partial removal of the AIS could trigger further melting of the AIS through a dynamic coupled ocean-atmosphere-sea ice feedback.

The feedbacks identified here are typically missing from recent LIG climate modelling studies[3]. Previous studies have identified a link between Southern Ocean meltwater forcing and subsurface warming due to a weakening of deep water formation[24]. Our study extends previous work by examining the combined temperature impacts of both ice removal and meltwater forcing, showing contrasting effects of each perturbation. Our results imply that 21st century climate model projections may be missing feedbacks that warm East Antarctica, in addition to meltwater-induced subsurface warming. This study high-lights the need to develop fully coupled climate model simulations that can interact dynamically with ice sheet changes, and thus capture feedbacks from melting parts of the AIS on the ocean-atmosphere circulation and temperature in the Antarctic region.

## Methods

### Model description

The experiments are carried out using the coupled climate model GFDL CM2.1[41]. The ocean component of the model uses a resolution of 1° latitude × 1° longitude × 50 levels, with a latitudinal grid refinement in the tropics down to 0.33° at the equator. The model also uses a tripolar grid, with a regular latitude-longitude grid south of 65°N, transitioning to a bipolar grid in the Arctic, with poles centred over Siberia and Canada[42]. The ocean model is modified from the original GFDL CM2.1 configuration to use MOM version 5.1.0, and uses a bottom roughness scheme for vertical mixing[43]. The sea ice model is the Sea Ice Simulator (SIS), which is a dynamical model with three vertical layers, one snow and two ice, with five ice thickness categories. It uses an elastic-viscous-plastic method to calculate ice stresses[44], and the thermodynamics use a modified Semtner three-layer scheme[45]. The sea ice component uses the same horizontal resolution as the ocean. The atmosphere compo-nent is the GFDL AM2, and uses a resolution of 2° latitude × 2.5° long-itude × 24 levels[41]. The land surface model is LM2, and is configured for PI conditions as in the original CM2.1 experiments. The land cover type is based on a historical land use distribution dataset, based on ref. 46. This classifies the land surface into 10 vegetation or land surface types[46], with corresponding properties of albedo, surface roughness, stomatal resistance, root depth and snow masking depth.

### Pre-Industrial and Last Interglacial experiments

The model is run in two configurations: a PI control run, and a LIG run. Both of these runs follow the experimental protocol of ref. 3, which specifies the greenhouse gas concentrations, orbital parameters and solar constant for both the PI and LIG simulations. These parameters are given in Table 1 of ref. 3. All vegetation types, soil types and runoff scheme are left unchanged for the LIG simulations and perturbation experiments (below). We keep the river runoff relocation map fixed regardless of changes to the ice sheets, since (i) the drainage-basin scale changes to river runoff are relatively small, and (ii) river runoff is a small signal compared to the meltwater perturbations, allowing us to attri-bute changes more directly to meltwater effects. The PI and LIG simu-lations are initialised from a modern observational dataset of temperature and salinity[47], and run for 1500 years. Both of these simulations reach a quasi-equilibrium with global mean SST increasing by 0.02–0.03 °C per century over the last 500 years. For the last 100 years of simulation, the temperature trend at 4000 m is between 0 and 0.03 °C per century in all simulations including the perturbation experiments. We account for this small drift in our analysis by treating the LIG simulation as a 'control' experiment, and subtracting it from the perturbation experiments at the same number of model years. The surface temperature changes found in the LIG compared with the PI simulation show similar seasonal changes (Supplementary Fig. S9) to the PMIP4 results[13] (their Fig. 5), and some differences in the annual mean response. As in the PMIP4 ensemble, our model simulates strong

northern hemisphere land warming in boreal summer (JJA) and land cooling in boreal winter (DJF) (Supplementary Fig. S9a, c). There is a contrasting signal of boreal summer cooling over tropical Africa and to a lesser extent India; these results are also present in the PMIP4 ensemble. Our results differ to PMIP4 in that we have an overall cooling in the annual mean, and a relative lack of Arctic warming, where the PMIP4 ensemble records an annual mean anomaly of around 2 °C. While PMIP4 finds an ensemble mean global anomaly of 0.5 °C, we find a global anomaly of −0.59 °C, similar to the lower end of the PMIP4 range (−0.5 to +2 °C). Below, we describe the perturbation experiments.

The MOC circulation in the PI and LIG runs is presented in Sup-plementary Fig. S10. The northern cell (AMOC) transports a similar volume in both simulations; 26.3 Sv and 26.1 Sv in the PI and LIG simulations respectively. The depth of the AMOC increases slightly, accounting for the minor changes in the streamfunction difference plot (Supplementary Fig. S10c). AABW formation decreases from 11.1 Sv in the PI to 9.8 Sv in the LIG. This decrease is evident in the positive difference around 70°S in Supplementary Fig. S10c. This decrease in AABW formation is also evident from a decrease in mixed layer depths in both the Ross and Weddell Seas (Supplementary Fig. S11). In the Weddell Sea, where AABW is predominantly formed, mixed layers are generally several hundred metres shallower, while in the Ross Sea, deep convection ceases completely (Supplementary Fig. S11).

### Partial Antarctic ice sheet removal experiments

Sea-level during the Last Interglacial was estimated to be between 1 and 9 m higher than present day[7]. Here we simulate the effects of this ice melt, using a perturbation derived from Golledge et al.[27], using their scenario of RCP4.5 forcing at year 5000, including the sub-grid basal melt interpolation (see their Fig. 1f; 'high' scenario). We chose this scenario, since it represents a 4.3 m SLE rise with respect to PI, i.e. in line with mid-range estimates of ice loss at the LIG. We use their modelled ice elevation as an indicative estimate of an Antarctic ice sheet reduced by 4.3 m, given sea level constraints from proxy information[5,8]. However, when comparing their ice sheet elevation for the historical period (1900 CE) with the control ice sheet in GFDL CM2.1 derived from observations but on a far coarser grid, there are some differences in baseline elevation (both higher and lower in dif-ferent regions). To account for these discrepancies in baseline ice sheet elevation, we applied a perturbation to the GFDL CM2.1 model in two ways: 1. using the anomaly of ice sheet elevation, taken as the difference between the RCP4.5 and 1900 CE simulations of Golledge et al.[27], and 2. using the absolute value of elevation in the RCP4.5 simulation of Golledge et al.[27]. In both cases, we made further adjustments so that the perturbed ice sheet was always less than or equal to the control (here the GFDL CM2.1 control) ice sheet elevation, and the anomalous elevation does not go below sea level.

This procedure yielded two estimates of ice loss: 1. Using the anomaly-derived perturbation, the resulting ice loss was 4.1 m SLE, and 2. using the absolute value of elevation, the resulting ice loss was 7.1 m SLE. For each of these values of total ice loss, we also perform a separate perturbation using the equivalent volume of freshwater forcing, and a combined ice loss and freshwater forcing experiment, as follows:

1. SL4.1: Removal of 4.1 m sea level-equivalent ice sheet using the RCP4.5 - 1900 CE anomaly from Golledge et al.[27] (Fig. 1a, c);
2. FW4.1: Freshwater forcing over 500 years adjacent to the Ant-arctic coast, with total volume equivalent to the ice loss derived in SL4.1;
3. COMB4.1: Combined perturbations of ice sheet removal (SL4.1), and freshwater forcing (FW4.1).
4. SL7.1: Removal of 7.1 m sea level-equivalent ice sheet using the RCP4.5 absolute ice elevation from Golledge et al.[27] (Fig. 1b, d);
5. FW7.1: Freshwater forcing over 500 years adjacent to the Ant-arctic coast, with total volume equivalent to the ice loss derived in SL7.1;

6. COMB7.1: Combined perturbations of ice sheet removal (SL7.1), and freshwater forcing (FW7.1).

All perturbations are started from year 1000 of the LIG experiment and using LIG orbital parameters and greenhouse gases. A summary table of these experiments is given in Supplementary Dataset 3.

We chose to derive perturbations from this model[27] because it is one of few publicly available Antarctic ice sheet reconstructions that matches the total sea-level change for the LIG. Several other ice modelling studies have also examined ice loss changes at the LIG[17,19]; these generally have a lower or similar total sea level change (3–4 m) than the scenario we have chosen. Part of our motivation for the choice of ref. 27 is to obtain a large signal of change (within the bounds of LIG sea-level change), in order to more clearly examine the mechanisms of climate responses, rather than finding the best possible LIG ice sheet reconstruction. In addition, the spatial pattern of AIS loss in these experiments is similar to previous simulations of the LIG Antarctic ice sheet[17–19]. Therefore, the AIS ice-mass loss forcing used here is consistent with modelling and paleo-proxy estimates.

We use the Antarctic bedrock map contained in ref. 27 to determine where the ice sheet overlies either marine or terrestrial points. Where the ice sheet is removed from marine locations, we replace the ice with ocean grid cells (formerly land grid cells in the model). Ocean grid cells are set to the depth of the marine bedrock, with an additional constraint that new ocean grid cells must have a minimum depth of 100 m. This minimum depth ensures numerical stability of the newly created shelf region. Where ice is removed from terrestrial locations, the land grid cells are lowered to the bedrock altitude[27]. In this case, the albedo and vegetation parameters are kept as the 'ice' type, to reflect the fact that the landscape is still predominantly ice- or snow-covered. Note that in the control run, the ocean grid is limited to 82°S, while in the SL4.1 configuration, the ocean grid is extended to 85°S (Fig. 1 and Supplementary Fig. S1).

### Freshwater forcing

We derive a freshwater forcing distribution based on the total ice lost between the pre-industrial and SL4.1 and SL7.1 ice sheets, converted to an equivalent freshwater volume. We spread this around Antarctica in binned 60° longitudinal sectors; where the total ice change in each sector is used to determine the total freshwater forcing for that sector (see Fig. 1a, b). The total freshwater volume is then spread around the coastline, in the 2 surface ocean grid cells adjacent to the nearest Antarctic land point; this distribution of freshwater forcing is shown in Supplementary Fig. S12. The rate of forcing is set according to a constant freshwater forcing period of 500 years, which amounts to 0.09 Sv and 0.16 Sv for the FW4.1 and FW7.1 experiments respectively. These rates are comparable to a maximal melting rate around 127 ka[19], but lower than a previous climate modelling study, which forced a 3.5 m SLE freshwater perturbation emulating a WAIS collapse in 100 years[22]. For the COMB4.1 and COMB7.1 experiments, we apply both the ice sheet removal and freshwater perturbations simultaneously. In these cases, there are more ocean grid cells present, especially south of 80°S along the Antarctic coast. Therefore the freshwater forcing region is somewhat larger in latitudinal extent (Supplementary Fig. S12), however the total flux remains the same as in the FW4.1 and FW7.1 experiments.

### Neutral density surfaces

For the calculation of $\omega$-surfaces of neutral density used in Figs. 3 and 4, we use the method of Stanley et al.[48]. The $\omega$-surfaces are generated from a 50-year average of temperature, salinity and pressure output from the model, which are then zonally averaged. The $\omega$-surfaces are generated at the reference location (49.5°S, 20.5°W) at depths of (100, 200, 500, 1000, 1500, 2000, 2500, 3000) m. These coordinates are in the Weddell Sea sector, but north of the zone of deep mixed layer depths, in order to be representative of the Southern Ocean

stratification but outside of the marginally stratified convection zone. Note that neutral surfaces are tangential to local isopycnal surfaces at their reference depth[48], representing pathways of along-isopycnal mixing and transport in the model.

## Data availability
The model data presented here are available at https://doi.org/10.5281/zenodo.10199378[49]. We thank Nicholas Golledge for making his ice sheet model output publicly available at https://osf.io/trwmc/.

## Code availability
The GFDL CM2.1 climate model code is publicly available at https://github.com/mom-ocean/MOM5, as part of the MOM5.1.0 release. We thank Geoff Stanley for providing code to calculate neutral surfaces, available at https://github.com/geoffstanley/neutralocean.

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

## Acknowledgements
This work was supported by Australian Research Council Grants FT180100606 (LM), DP180100048 (KJM and LM), DE220100279 (DKH) and SR200100008 (LM). The simulations were undertaken with the assistance of resources and services from the National Computational Infrastructure (NCI), which is supported by the Australian Government.

## Author contributions
D.K.H. and L.M. led the study and designed the experiments, with advice from K.J.M. L.M. and K.J.M. shaped the literature review and context of the study. A.M.H. advised on Antarctic ocean circulation and ice removal experimental design. D.K.H. ran the experiments and wrote the first draft of the manuscript. All authors contributed to the writing and editing of the manuscript.

## Competing interests
The authors declare no competing interests.
