## [Peer Review File · Nature Communications]

East Antarctic warming forced by ice loss during the Last InterglacialReviewer #1 (Remarks to the Author):

Review of : "East Antarctic warming triggered by West Antarctic ice loss during the Last Interglacial"

Hutchinson et al. investigate potential feedbacks of Last Interglacial (LIG) Antarctic Ice Sheet (AIS) retreat acting on the atmospheric and oceanic circulation. Their motivation is the apparent discrepancy between modelled LIG climate conditions (e.g. in PMIP4) and the actual observed state revealed by ice core proxy data (surface air temperatures, SAT) and marine sediment cores (sea surface temperatures, SST). This discrepancy is a long-standing issue in the paleo-modelling community and very plausibly caused by missing ice-atmosphere-ocean feedbacks in the relatively coarse resolution PMIP4 models. Fully coupled climate models including interactive and evolving ice sheets are still difficult to realize due to prohibitive computational constraints and technical challenges such as evolving ice shelf cavities. To overcome these limitations ice sheet boundary conditions can be adjusted to the respective climate epoch and freshwater hosing can mimic increased ice discharge e.g. in Interglacials. This is the approach Hutchinson et al. take in this work. Their main finding is, that a 500 year freshwater hosing during the LIG, sourced from the West Antarctic Ice Sheet (WAIS) and combined with a modified ice sheet topography (partial WAIS collapse, including bathymetric changes), has considerable effects on the Southern Ocean thermal regime as well as on SATs over the AIS. The LIG SAT anomalies found in Hutchinson's model simulations are closer to ice core reconstructions compared to the results of the PMIP4 ensemble. The subsurface ocean temperature anomalies the authors model here are of sufficient magnitude to affect grounding line dynamics during the LIG. I congratulate the authors on an interesting piece of work. Overall, this is an interesting and well written study focused on a highly relevant research question in earth system science. It could be a valuable contribution; however, I have several comments on how the author present their findings. Additionally, I find parts of the methodological approach to be inconsistent. To make this a viable contribution to Nature Comm. in my opinion major changes to the manuscript and potentially additional experiments are necessary. While I certainly hope so, I leave it to the editor and the authors to decide whether this is possible. In the following I will outline my main comments followed by some smaller comments.

Major points:

- 1. Hutchinson et al. write in their conclusions, that their's is the first study "demonstrating ... the contrasting climatic impacts of Antarctic ice removal and Southern Ocean meltwater input". This sentence needs moderation as to my knowledge several studies (e.g. Holloway et al. 2016 in Nature Communications) follow similar experimental procedures albeit with different foci in their analysis. When reading the manuscript by Hutchinson et al., I was missing a discussion how their findings differ from previous studies and what novel methodologies they apply. This should be expanded on, certainly in the methods section, but also in the main manuscript.**
- 2. The title claims that the climate response to changes of WAIS is investigated here. However, the authors write that they consider a scenario akin to 4.5 m AIS SLE volume change during the LIG which is inconsistent. Should the whole WAIS collapse, global sea levels would rise by "only" 3-3.5 m (see e.g. Bamber et al. 2009). In light of the "4.5 m scenario" this would imply at least some East Antarctic Ice Sheet (EAIS) contributions. Furthermore, the boundary conditions used here clearly show only partial WAIS collapse (e.g. Figure 1) thus the EAIS contribution must be considerable.**
- 3. This brings me to the last main point. The experimental setup is designed to focus on freshwater hosing from the WAIS, specifically from the Amundsen/Bellinghousen Sea embayment as well as the Ross and Weddell Sea. This is inconsistent with the**

topographic boundary conditions applied: Figure 1 shows that e.g. the Recovery Basin glaciers are clearly retreated. The Recovery Basin is technically East Antarctica but drains but the freshwater from ice sheet retreat would end up in the Weddell Sea, so that's consistent with the hosing approach. However, large swathes of the coastline of Dronning Maud Land are gone and Wilkes and Aurora Basin Glaciers have retreated. For the sake of consistency, freshwater should be channeled through these regions as well. Additional experiments including freshwater hosing from the EAIS, and analysis of the impact of regional hosing on the different ocean sectors would greatly strengthen this study (see below).

Specific Comments:

I think there are several places in the manuscript which would benefit from references e.g. L36, L48, L50 here e.g. Sutter et al. 2016, Pollard et al. 2016. L51 e.g. Golledge et al. 2019

L59: Figure 1 shows the ice sheet representation on the GCM-grid resolution. You mention in the methods that you take the ice sheet model output from the RCP4.5 simulation at the year 5000 in Golledge et al. 2015 and then regrid it to the native GCM-resolution? This means (as one can see in Figure 1) that you not only remove parts of the WAIS but also the EAIS (e.g. in the Recovery Basin, Wilkes subglacial basin and Aurora basin, coastline of Dronning Maud Land exhibits drastic changes as well). This means you can't strictly assign the climatic changes to WAIS topographic LIG-responses exclusively. I suggest you discuss this in the manuscript and potentially adjust the framing accordingly. Furthermore, looking at Figure 1 in Golledge et al. 2015 from which you take the boundary conditions, it seems that the WAIS configuration in your GCM boundary conditions is the "low scenario" i.e. 2.77 m SLE, is that correct? In that case, most of WAIS seems to be still intact in your representation, which means that the majority of sea level equivalent ice volume change has to come from the East Antarctic Ice Sheet. A complete loss of WAIS amounts to ca. 3 m sea level rise (e.g. Bamber et al. 2009). You state that yours is a configuration with 4.5m sea level rise but only partial WAIS collapse. Please elaborate and expand the discussion/caveats of your approach accordingly.

Figure 2: I suggest you include place names in figure 1 as an orientation for the reader unfamiliar with Southern Ocean sector names. Also, consider including an elevation map from the original ice sheet model showing the ice sheet thk changes compared to the ctrl (e.g. with a diverging RdBu cmap).

L115 surface cooling alone

L118 I would argue that this is a very striking finding as subsurface ocean temperature ultimately drive ice sheet retreat/advance (the grounding line does not "see" SST). In general, the effect of AIS changes on the subsurface ocean conditions should be highlighted and discussed more in the manuscript as it reveals a positive feedback if including freshwater forcing (as e.g. shown and discussed in Golledge et al. 2019, Bronselaer et al. 2018) and I think goes beyond what e.g. Holloway et al. 2016 discussed. However, I think that additional experiments would be necessary as the boundary conditions you apply clearly exhibit substantial EAIS changes as well. How would these affect ocean circulation? Would your results change (e.g. the large Ross Sea ocean warming) if you would add a freshwater feedback in East Antarctica as well? What mechanisms would be in place then?

L161 again, this statement is incomplete, as you also include EAIS topographic changes (but only WAIS freshwater forcing, which is inconsistent). Furthermore, the largest feedback of WAIS retreat seems to be a substantial subsurface warming of the Ross Sea.

L166 e.g. Holloway et al. 2016 conduct similar experiments also focused on the LIG.

L190 Ice core constrained ice sheet modelling show's that this (Wilkes Subglacial Basin) sector very likely remained stable during the LIG (Sutter et al. 2020). Uranium isotopes in subglacial precipitates also hint at LIG ice sheet stability in the Wilkes Subglacial Basin (Blackburn et al. 2020). Furthermore, Sutter et al. 2016 show sea level rise of ca. 3-4 m (mid range) for the LIG without invoking MICI.

L197 this sentence appears to be incomplete?

L198 see e.g. Holloway et al. 2016 and other studies investigating the climate response to ice sheet topographic changes and freshwater hosing. In general, I suggest to significantly expand the method section by detailing how your experimental setup goes beyond/differs from previous studies.

L253 there are at least two studies explicitly simulating the AIS during the LIG (Sutter et al. 2016, DeConto et al. 2016). Furthermore, Golledge et al. 2019 discusses the effect of changes in ocean stratification due to freshwater release from the AIS.

L255- No, see below.

L266- not consistent with the physical constraints. 1. WAIS only amounts to ca. 3 m SLE ice volume (e.g. Bamber et al. 2009), 2. only partial collapse of WAIS in boundary conditions, -> substantial contr. from EAIS.

Bern, 05.07.2022

Johannes Sutter

Reviewer #2 (Remarks to the Author):

This paper presents modelling attempt to illustrate the impacts of Antarctic-ice partial removal and the associated Southern Ocean meltwater input on the Antarctic climate change in the LIG with respect to the PI. New to existing studies, their ESM modelling results have suggested the contrasting climatic impacts of Antarctic ice removal and Southern Ocean meltwater input on the spatial pattern of the LIG Antarctic surface air temperature during the LIG. This work is of significance to the field of paleoclimatology and has implications for present day global warming. However, in the manuscript, both the modelling set-ups and climatic mechanism remain to be further clarified. I would encourage a re-submission of a revised manuscript.

Comments:

1. In 6.3, the SL4.5 experiment set-up is discribed that: Removal of 4.3 m (here it should be 4.5?) sea level-equivalent ice sheet from Antarctic topography and adjustment of ocean bathymetry. So, I would expect that the entire topography of the Antarctic ice and the coatal line in the LIG are changed. However, in the manuscript, the mechanism is only attributed to the West Antarctic ice loss, e.g. as mentioned in the title. How to undersrtand this?

2. In Fig.1, a figure panel for LIG-to-PI topography anomolies will be useful to understand the ice change

3. l.234-235, 'The surface temperature changes found in the LIG compared with the PI simulation are broadly in agreement with PMIP4 results5 (Figure S7)'. Acturally, this is not true. The Figure S7 is quite different from Fig. 5 in ref.5.

4. l.227, How to understand 'the runoff scheme are left unchanged'? E.g. river freshwater discharge amount, river mouth location.

5. l.44-45, 'implies a larger sea level rise by 2100 than many process-based studies', how to understand by 2100?

6. l.54-55, 'the stability' is not discussed in this work, but a shift of mean climate state.

7. I.62-63, The Weddell-Sea sea ice edge shows W-S opposite pattern in sea ice edge change, rather than a common retreat.
8. I.64, sea ice edge around most of East Antarctica is actually reduced, not steady, see Fig. 2a.
9. I.67-68, 100-500 m depth in the the Ross Sea becomes colder rather than warmer, see Fig. 2d.
10. I.69-72, according to Fig. S2a, an anti-clockwise atmosphere circulation over the Amundsen, Bellingshausen and Ross Seas is shown. It seems as a key pattern to explain the regional climate. Is it related to the Land-sea-mask change, e.g. enlarged Ross-Sea bay due to coastal line retreat?
11. I.101-102, 'First, the weakening of the equator-to-pole gradient (due to Antarctic surface warming) drives a weakening of the midlatitude westerlies', by what mechanism?
12. I.105-106, Fig. S5 is not able to present 'The southward shift of the easterlies also strengthens the Antarctic Slope Current (ASC; Figure S5).', please add a figure for anomalies.
13. I.119-120, 'and by 0.5-1 C along most of the West Antarctic coast (180E - 60W, Figure 2e).', Fig.2e is not able to present this change.
14. I.129-131, 'The strength of midlatitude westerly winds increases in the FW4.5 experiment, likely due to an enhanced equator-to-pole gradient, contrary to the circulation changes in the ice loss experiment.' Similarly, by what mechanism?
15. I.139-140, 'since the ice elevation does not change at these proxy locations.' Same as comment #2, a figure panel for LIG-to-PI topography anomalies is requested.
16. I.147-149, 'On the other hand, a small SST increase is simulated along the East Antarctic coast, which may be due to the minor retreat in the land ice edge in this region (Figure 1).' Does this mean land-ice grid becomes ocean grid cells in the model? Please clarify.
17. I.158-159, 'is driven by a southward expansion of CDW below the freshwater layer' Why to exclude the impact of reduced AABW formation. Any modelling signal to indicate this?
18. In general, how deep is the ocean channel between the WAIS Island and Antarctic main land, when the ice is reduced? Then, how is the water exchange between the Weddell Sea and Amundsen Sea? This is also key to understand the full process.
19. Please add the information about the MOC anomalies between LIG and PI. Also important to understand the PI-to-LIG ocean change.

Reviewer #3 (Remarks to the Author):

Review of NCOMMS-22-16471-T: " East Antarctic warming triggered by West Antarctic ice loss during the Last Interglacial".

In this paper the authors discuss the impact of both WAIS removal and freshwater input on the climate and ocean state in the southern Hemisphere during the LIG. They use a climate model with a prescribed ice sheet and freshwater forcing. One issue they investigate is if the mismatch between the LIG127ka PMIP4 simulations and southern hemisphere proxy records (SAT and SST) was due to the choice of an PI AIS in the model setup. The authors focus on indefinitely changes in southern hemisphere climate state specially from ice-sheet driven feedbacks (changes in bathymetry/elevation) from those related to freshwater by running three climate simulations.

The paper is well written, and the results are clearly presented. However, I would the authors to expand on previous publications which have investigate the combination of WAIS and FWF for the LIG, although they may be from older generation of climate models. For example, Holloway et al., 2016 also performed climate model simulations with a reduced AIS and FWF.(I was not associated with this study). I would recommend for publication with the minor changes below.

Section 2: Question about lapse rate correction: I would like more details on the

reasoning of this correction for the final COMB4.5 results (shown in the paper). The results in Fig3 are the SAT with the lapse-rate correction, so removing any warming signal associated with a change in altitude due to the retreat of the WAIS. However, as the aim of the SL4.5 is to investigate the impact of a reduced WAIS, this correction removes/reduces this impact. Why are the corrected results not shown in the SOM? I am not sure of the rationale? Is this to make the results more consistent with FW4.5?

Minor text corrections:

Line 35: "Coupling a dynamic ice sheet with ocean-atmosphere-sea-ice-land ... remains a significant technical barrier". There have been simulations with dynamic ice sheet (Sommers et al, 2021). To be clearer, I would state a dynamic Antarctic ice sheet. This is a challenge due to the ice-ocean coupling required for a large marine based ice sheet.

Results section2: SL4.5 experiment compared to COMB4.5

From the SL4.5 (Fig2d) there is a pronounced warming signal ($> 2C$) at 100-500m between 140W and 180W. What is the origin of this? It does not occur when running the COMB4.5 experiment, unlike most other signals (Wind speed, SAT)? Is there a 2nd feedback so the FWF dominates the response in the ocean?

Discussion:

Line 160 (related to FigS6)" Moreover, the East Antarctic subsurface anomaly is more positive than in FW4.5, which partly reflects the warming caused by the continental ice sheet reduction" It could be the colour scale but comparing Fig.2e and f most of EAIS appear to be surrounded by 'white' region, so a very similar temperature as in FW4.5 (so colder), apart from close to the Wilkes Basin. Also, in FigS6, with deeper SST anomalies the ice sheet only simulation (FigS6d), there is a strong cooling along the coast. Line 182:"there are also areas for East Antarctica' where reverse-sloping bedrock...' Can you mention which of these three regions coincide with simulated warming/cooling anomalies. It is interesting the large warming anomaly near the Wilkes SG Basin, but the Amery is associated with a cooling.

Figures:

Figure 2-3. Could you increase the latitude extent to be consistent with SOM plots. The lateral extent of the warmer anomalies (i.e Fig.2d) is chopped off.

Figure 3: what are the circles? Are there the ice core sites? Are the anomalies larger than 5C? if so can you edit the scale bar to indicate this?

Methods:

Table: Model setup: could the authors add a table in the SOM of model set up (forcings), names; ice sheets (small/large); size of FWF etc. Just as summary for the model setup and boundary conditions.

Table: Proxy data: Throughout the text the authors compare the SAT/SST from the simulations to proxy data and other values from PMIP4 simulations. Can these be summarised in a table. For example, the 4 proxy ice core records (line 80); SST temperature records (line 25-26). Can the authors compare to other proxy records, for example those shown in the Otto-Bliesner et al., 2021, summary PMIP4 LIG127ka paper.

References:

Holloway, M., Sime, L., Singarayer, J. et al. Antarctic last interglacial isotope peak in response to sea ice retreat not ice-sheet collapse. *Nat Commun* 7, 12293 (2016). <https://www.nature.com/articles/ncomms12293>

Otto-Bliesner et al., 2021 Large-scale features of Last Interglacial climate: results from evaluating the lig127k simulations for the Coupled Model Intercomparison Project

(CMIP6)–Paleoclimate Modeling Intercomparison Project (PMIP4)

<https://cp.copernicus.org/articles/17/63/2021/>

We thank the reviewers for their thoughtful and constructive comments, which have helped to greatly improve the manuscript. Below, we have copied the reviewers' comments in black text, and provide our responses in blue text.

Kind Regards,
David Hutchinson

Reviewer #1 (Remarks to the Author):

Review of : "East Antarctic warming triggered by West Antarctic ice loss during the Last Interglacial"

Hutchinson et al. investigate potential feedbacks of Last Interglacial (LIG) Antarctic Ice Sheet (AIS) retreat acting on the atmospheric and oceanic circulation. Their motivation is the apparent discrepancy between modelled LIG climate conditions (e.g. in PMIP4) and the actual observed state revealed by ice core proxy data (surface air temperatures, SAT) and marine sediment cores (sea surface temperatures, SST). This discrepancy is a long-standing issue in the paleo-modelling community and very plausibly caused by missing ice-atmosphere-ocean feedbacks in the relatively coarse resolution PMIP4 models. Fully coupled climate models including interactive and evolving ice sheets are still difficult to realize due to prohibitive computational constraints and technical challenges such as evolving ice shelf cavities. To overcome these limitations ice sheet boundary conditions can be adjusted to the respective climate epoch and freshwater hosing can mimic increased ice discharge e.g. in

Interglacials. This is the approach Hutchinson et al. take in this work. Their main finding is, that a 500 year freshwater hosing during the LIG, sourced from the West Antarctic Ice Sheet (WAIS) and combined with a modified ice sheet topography (partial WAIS collapse, including bathymetric changes), has considerable effects on the Southern Ocean thermal regime as well as on SATs over the AIS. The LIG SAT anomalies found in Hutchinson's model simulations are closer to ice core reconstructions compared to the results of the PMIP4 ensemble. The subsurface ocean temperature anomalies the authors model here are of sufficient magnitude to affect grounding line dynamics during the LIG. I congratulate the authors on an interesting piece of work. Overall, this is an interesting and well written study focused on a highly relevant research question in earth system science. It could be a valuable contribution; however, I have several comments on how the author present their findings.

Additionally, I find parts of the methodological approach to be inconsistent. To make this a viable contribution to Nature Comm. in my opinion major changes to the manuscript and potentially additional experiments are necessary. While I certainly hope so, I leave it to the editor and the authors to decide whether this is possible. In the following I will outline my main comments followed by some smaller comments.

We thank Dr. Sutter for his constructive and thoughtful comments, and positive encouragement for resubmission. We have made substantial revisions to address the major concerns, including revising and re-running all but one of the perturbation

experiments to achieve better consistency between the ice sheet and meltwater forcing, and more thoroughly addressing previous literature. We have also expanded our experimental scope to include a smaller and a larger perturbation of ice (4.1 m SLE and 7.1 m SLE), with equivalent amounts of freshwater forcing in each case. Importantly, we have taken a more careful approach exactly describe the ice perturbations and match the location of freshwater forcing around the Antarctic coast to align with where the ice has been removed. We believe that the new experimental design and updated manuscript thoroughly address Dr Sutter's concerns.

Major points:

1. Hutchinson et al. write in their conclusions, that their's is the first study "demonstrating ... the contrasting climatic impacts of Antarctic ice removal and Southern Ocean meltwater input". This sentence needs moderation as to my knowledge several studies (e.g. Holloway et al. 2016 in Nature Communications) follow similar experimental procedures albeit with different foci in their analysis. When reading the manuscript by Hutchinson et al., I was missing a discussion how their findings differ from previous studies and what novel methodologies they apply. This should be expanded on, certainly in the methods section, but also in the main manuscript.

We have now included at several points in the Introduction and Discussion a comparison with Holloway et al (2016), and added further references covering previous work on LIG ice sheet modelling and observations. We acknowledge that there are some similarities to Holloway et al's work, but also several differences in our methods, and importantly, we achieve our best fit to the LIG proxies in East Antarctica for different reasons than their study (which argued sea ice reduction was the most important factor).

2. The title claims that the climate response to changes of WAIS is investigated here. However, the authors write that they consider a scenario akin to 4.5 m AIS SLE volume change during the LIG which is inconsistent. Should the whole WAIS collapse, global sea levels would rise by "only" 3-3.5 m (see e.g. Bamber et al. 2009). In light of the "4.5 m scenario" this would imply at least some East Antarctic Ice Sheet (EAIS) contributions. Furthermore, the boundary conditions used here clearly show only partial WAIS collapse (e.g. Figure 1) thus the EAIS contribution must be considerable.

First, we agree that our original description of the WAIS component of the perturbation was incorrect and we thank Dr. Sutter for catching this. There was indeed a considerable component of EAIS change in our experiments, and we have reframed the manuscript to address this. Second, in the process of revising the manuscript, we realised that there were some unintended perturbations to East Antarctica that we had not scrutinised properly.

Our control simulation is forced with an imposed ice sheet based on present day observations while the sensitivity experiments in our first submission were forced with ice elevations directly from the Gollledge et al (2015) RCP4.5 ice sheet model

scenario. This was an inconsistent approach, because the Golledge et al simulations started with a simulated, and therefore slightly different, control ice sheet. This resulted in differences between the ice sheets used for our sensitivity runs and the control run that were not only due to climate forcing, but also to the inherent difference between the two control ice sheets. As a result, some regions of the East Antarctic ice sheet were lower because of the mismatch of the control ice sheets.

In the revised manuscript, we have corrected this inconsistency by calculating the difference between Golledge et al's RCP4.5 simulation and their year 1900 control run and then adding this "anomaly" in ice sheet elevation to our control run ice sheet. This enabled us to create a perturbation that was more in line with their simulated warming effects; such that the ice removal was more focused on marine ice sheets and glacial valleys where melting is likely to occur. This perturbation is labelled SL4.1, representing a 4.1 m SLE change in ice.

In addition, we decided to retain our original ice sheet perturbation (previously called SL4.5), since it produced interesting results. And, after properly accounting for the anomalies in ice elevation, we have now renamed this experiment SL7.1, since in our grid configuration, the total ice volume change is in fact 7.1 m SLE.

The new perturbations are presented more clearly in our new Figure 1, with ice elevation anomaly fields as requested. The new SL4.1 and SL7.1 ice elevation anomalies are shown in Figure 1a,b, while the change in bathymetry is shown in Figure 1c,d. The absolute values of elevation and bathymetry are shown in Supplementary Figure S1. We now also quantify the sea level equivalent (SLE) ice sheet change for each 60 deg sector of Antarctica. Furthermore, we quantify the contributions of the WAIS, EAIS and Antarctic Peninsula sectors according to the drainage basin definition of each sector, presented in Supplementary Table 1.

3. This brings me to the last main point. The experimental setup is designed to focus on freshwater hosing from the WAIS, specifically from the Amundsen/Bellinghousen Sea embayment as well as the Ross and Weddell Sea. This is inconsistent with the topographic boundary conditions applied: Figure 1 shows that e.g. the Recovery Basin glaciers are clearly retreated. The Recovery Basin is technically East Antarctica but drains but the freshwater from ice sheet retreat would end up in the Weddell Sea, so that's consistent with the hosing approach. However, large swathes of the coastline of Dronning Maud Land are gone and Wilkes and Aurora Basin Glaciers have retreated. For the sake of consistency, freshwater should be channeled through these regions as well. Additional experiments including freshwater hosing from the EAIS, and analysis of the impact of regional hosing on the different ocean sectors would greatly strengthen this study (see below).

We agree that the runoff in the original experiments was not consistent with the distribution of ice sheet changes. We therefore redesigned the runoff to be more widely distributed around Antarctica. We used the 60 deg sector-by-sector values noted above (Figure 1a,b) to define the magnitude of freshwater forcing for each sector in each ice loss scenario, in order to achieve consistency between the ice sheet changes and the freshwater forcing.

Specific Comments:

I think there are several places in the manuscript which would benefit from references e.g.

L36, L48, L50 here e.g. Sutter et al. 2016, Pollard et al. 2016. L51 e.g. Golledge et al. 2019

These references have been added accordingly.

L59: Figure 1 shows the ice sheet representation on the GCM-grid resolution. You mention in the methods that you take the ice sheet model output from the RCP4.5 simulation at the year 5000 in Golledge et al. 2015 and then regrid it to the native GCM-resolution? This means (as one can see in Figure 1) that you not only remove parts of the WAIS but also the EAIS (e.g. in the Recovery Basin, Wilkes subglacial basin and Aurora basin, coastline of Dronning Maud Land exhibits drastic changes as well). This means you can't strictly assign the climatic changes to WAIS topographic LIG-responses exclusively. I suggest you discuss this in the manuscript and potentially adjust the framing accordingly. Furthermore, looking at Figure 1 in Golledge et al. 2015 from which you take the boundary conditions, it seems that the WAIS configuration in your GCM boundary conditions is the "low scenario" i.e. 2.77 m SLE, is that correct? In that case, most of WAIS seems to be still intact in your representation, which means that the majority of sea level equivalent ice volume change has to come from the East Antarctic Ice Sheet. A complete loss of WAIS amounts to ca. 3 m sea level rise (e.g. Bamber et al. 2009). You state that yours is a configuration with 4.5m sea level rise but only partial WAIS collapse. Please elaborate and expand the discussion/caveats of your approach accordingly.

For clarification, we took our boundary conditions from the scenario shown in Golledge et al (2015) Figure 1f, using the 'high' scenario, which includes a sub-grid basal melting parameterisation. This entails a SLE change of 4.30 m, rather than 2.77 m for the 'low' scenario. On the issue of WAIS contribution, our revised perturbation is indeed less than the 3 m estimated by Bamber et al (2009), and we have quantified this amount more clearly in the Supplementary material. The WAIS component of ice loss is 2.05 m and 2.28 m SLE in SL4.1 and SL7.1 respectively.

Figure 2: I suggest you include place names in figure 1 as an orientation for the reader unfamiliar with Southern Ocean sector names. Also, consider including an elevation map from the original ice sheet model showing the ice sheet thk changes compared to the ctrl (e.g. with a diverging RdBu cmap).

We have included some place names in Figure 1 as suggested, and included a red-blue diverging colour depiction of the ice sheet perturbation compared with the control. We have also noted on the difference plot the 60 deg sector-by-sector ice perturbations as discussed above.

L115 surface cooling alone

This has been added.

L118 I would argue that this is a very striking finding as subsurface ocean temperature ultimately drive ice sheet retreat/advance (the grounding line does not "see" SST). In general, the effect of AIS changes on the subsurface ocean conditions should be highlighted and discussed more in the manuscript as it reveals a positive feedback if including freshwater forcing (as e.g. shown and discussed in Golledge et al. 2019, Bronselaer et al. 2018) and I think goes beyond what e.g. Holloway et al. 2016 discussed. However, I think that additional experiments would be necessary as the boundary conditions you apply clearly exhibit substantial EAIS changes as well. How would these affect ocean circulation? Would your results change (e.g. the large Ross Sea ocean warming) if you would add a freshwater feedback in East Antarctica as well? What mechanisms would be in place then?

We have revised and re-run the perturbation experiments so that there is now a component of freshwater forcing along the coast of East Antarctica, consistent with the ice sheet changes in that region. The results are modified, but still show many of the same features as the original experiments; namely a significant subsurface warming in the Ross Sea, and deep sea changes that align with changes to the rate of AABW formation and flattening of isoneutral slopes.

L161 again, this statement is incomplete, as you also include EAIS topographic changes (but only WAIS freshwater forcing, which is inconsistent). Furthermore, the largest feedback of WAIS retreat seems to be a substantial subsurface warming of the Ross Sea.

We have revised the EAIS/WAIS discussion accordingly and highlight the subsurface warming of the Ross Sea.

L166 e.g. Holloway et al. 2016 conduct similar experiments also focused on the LIG.

As discussed above, we now include comparisons to Holloway et al (2016).

L190 Ice core constrained ice sheet modelling show's that this (Wilkes Subglacial Basin) sector very likely remained stable during the LIG (Sutter et al. 2020). Uranium isotopes in subglacial precipitates also hint at LIG ice sheet stability in the Wilkes Subglacial Basin (Blackburn et al. 2020). Furthermore, Sutter et al. 2016 show sea level rise of ca. 3-4 m (mid-range) for the LIG without invoking MICI.

We now acknowledge constraints provided in Sutter et al (2020) and Blackburn et al (2020) which suggest ice sheet melting in the Wilkes Subglacial Basin was less than 0.8 m SLE. Our ice sheet perturbations in this region are broadly consistent with this constraint: in the SL4.1 experiment there is only 0.12 m SLE ice loss between 120 and 180 °E, while in SL7.1 experiment there is 1.14 m between 120 and 180 °E, i.e. covering a larger region than the Wilkes Subglacial Basin.

L197 this sentence appears to be incomplete?

The sentence has been revised.

L198 see e.g. Holloway et al. 2016 and other studies investigating the climate response to ice sheet topographic changes and freshwater hosing. In general, I suggest to significantly expand the method section by detailing how your experimental setup goes beyond/differs from previous studies.

Agreed, we have expanded our discussion of previous work including Holloway et al (2016).

L253 there are at least two studies explicitly simulating the AIS during the LIG (Sutter et al. 2016, DeConto et al. 2016). Furthermore, Golledge et al. 2019 discusses the effect of changes in ocean stratification due to freshwater release from the AIS.

We have added further references to Sutter et al (2016), DeConto and Pollard (2016) and Golledge et al (2019) on this subject.

L255- No, see below.

This section has been revised.

L266- not consistent with the physical constraints. 1. WAIS only amounts to ca. 3 m SLE ice volume (e.g. Bamber et al. 2009), 2. only partial collapse of WAIS in boundary conditions, -> substantial contr. from EAIS.

We now clearly delineate contributions from the WAIS and EAIS, as discussed previously.

Bern, 05.07.2022

Johannes Sutter

Reviewer #2 (Remarks to the Author):

This paper presents modelling attempt to illustrate the impacts of Antarctic-ice partial removal and the associated Southern Ocean meltwater input on the Antarctic climate change in the LIG with respect to the PI. New to existing studies, their ESM modelling results have suggested the contrasting climatic impacts of Antarctic ice removal and Southern Ocean meltwater input on the spatial pattern of the LIG Antarctic surface air temperature during the LIG. This work is of significance to the field of paleoclimatology and has implications for present day global warming. However, in the manuscript, both the modelling set-ups and climatic mechanism remain to be further clarified. I would encourage a re-submission of a revised manuscript.

We thank the reviewer for their constructive comments, and we have revised the manuscript accordingly.

Comments:

1. In 6.3, the SL4.5 experiment set-up is described that: Removal of 4.3 m (here it should be 4.5?) sea level-equivalent ice sheet from Antarctic topography and adjustment of ocean bathymetry. So, I would expect that the entire topography of the Antarctic ice and the coastal line in the LIG are changed. However, in the manuscript, the mechanism is only attributed to the West Antarctic ice loss, e.g. as mentioned in the title. How to understand this?

As mentioned in response to Reviewer 1, we have now redesigned the perturbation experiments and conducted new simulations to correct some inconsistencies and apply freshwater forcing around the East Antarctic coast in line with the upstream ice sheet changes. We now clearly quantify how much ice is removed in each sector of Antarctica (Figure 1a,b), and give a breakdown of the WAIS, EAIS and Antarctic Peninsula components.

2. In Fig.1, a figure panel for LIG-to-PI topography anomalies will be useful to understand the ice change

These anomalies are now presented in Figure 1.

3. I.234-235, 'The surface temperature changes found in the LIG compared with the PI simulation are broadly in agreement with PMIP4 results5 (Figure S7).' Actually, this is not true. The Figure S7 is quite different from Fig. 5 in ref.5.

We have revised this statement to emphasise similarities in the seasonal patterns with as in PMIP4, while noting differences in the annual mean response and in particular regions such as the Arctic.

4. I.227, How to understand 'the runoff scheme are left unchanged'? E.g. river freshwater discharge amount, river mouth location.

We chose to leave the river runoff location the same in all experiments, since (i) the basin scale changes to river runoff are relatively small, and (ii) river runoff is a small signal compared to the meltwater perturbations, allowing us to attribute changes more directly to meltwater effects.

5. I.44-45, 'implies a larger sea level rise by 2100 than many process-based studies', how to understand by 2100?

This sentence has been rephrased; we mainly highlight that alternative approaches to ice sheet modelling still predict significant reductions in the WAIS during the 21st century without a MICI parameterisation.

6. I.54-55, 'the stability' is not discussed in this work, but a shift of mean climate state.

Here we have clarified that our study investigates impacts on climate in regions that affect the stability of the remaining ice sheet.

7. I.62-63, The Weddell-Sea sea ice edge shows W-S opposite pattern in sea ice edge change, rather than a common retreat.

We now acknowledge that an expansion of sea ice occurs in the Weddell Sea, and since the experimental design has changed, the results have been rewritten accordingly.

8. I.64, sea ice edge around most of East Antarctica is actually reduced, not steady, see Fig. 2a.

We have now rewritten this sentence to distinguish between regions with sea ice retreat (0-60°E), and sea ice relatively unchanged (60-180°E). We also note here that the data and Figure have changed.

9. I.67-68, 100-500 m depth in the Ross Sea becomes colder rather than warmer, see Fig. 2d.

The cooling identified by the reviewer is a neutral signal in the new perturbation experiment. We have revised this sentence to describe the new results. We have also re-organised our figures to focus more on the zonal mean temperature sections, which better illustrate the flattening of isoneutral slopes that ultimately drive the subsurface warming (Figure 4, 5), while the 100-500 m subsurface temperature anomalies are now in the supplementary material (Figure S2).

10. I.69-72, according to Fig. S2a, an anti-clockwise atmosphere circulation over the Amundsen, Bellingshausen and Ross Seas is shown. It seems as a key pattern to explain the regional climate. Is it related to the Land-sea-mask change, e.g. enlarged Ross-Sea bay due to coastal line retreat?

Since we have updated the perturbation experiments and the figure, we have updated our explanation of this result. We now characterise the change more as a weakening of the midlatitude westerly winds than an anti-clockwise circulation and have added the control values of wind to clarify this point (Figure S6a). We refer to polar surface warming driving a decrease in baroclinic instability, thus weakening the westerly winds, as in Butler et al (2010).

11. I.101-102, 'First, the weakening of the equator-to-pole gradient (due to Antarctic surface warming) drives a weakening of the midlatitude westerlies', by what mechanism?

We have added the following reasoning for this weakening: "polar warming at the surface has been found to reduce baroclinic instability and therefore weaken the midlatitude westerlies (Butler et al, 2010)".

12. I.105-106, Fig. S5 is not able to present 'The southward shift of the easterlies

also strengthens the Antarctic Slope Current (ASC; Figure S5).’, please add a figure for anomalies.

We have added an anomaly plot for the surface ocean velocities.

13. I.119-120, ‘and by 0.5-1 C along most of the West Antarctic coast (180E - 60W, Figure 2e).’, Fig.2e is not able to present this change.

This figure has been revised, and that statement has been removed.

14. I.129-131, ‘The strength of midlatitude westerly winds increases in the FW4.5 experiment, likely due to an enhanced equator-to-pole gradient, contrary to the circulation changes in the ice loss experiment.’ Similarly, by what mechanism?

As noted above, we have revised our discussion of westerly wind changes to include a reference to polar warming (in isolation) causing a reduction in baroclinic instability and westerly winds (Butler et al, 2010). This case is the converse: polar surface cooling causing an increase in westerly winds.

15. I.139-140, ‘since the ice elevation does not change at these proxy locations.’ Same as comment #2, a figure panel for LIG-to-PI topography anomalies is requested.

This statement has been rewritten the new perturbation experiments. We now present topography anomaly plots in Figure 1 as suggested.

16. I.147-149, ‘On the other hand, a small SST increase is simulated along the East Antarctic coast, which may be due to the minor retreat in the land ice edge in this region (Figure 1).’ Does this mean land-ice grid becomes ocean grid cells in the model? Please clarify.

Yes, we modify the land-sea mask to include more ocean cells which were previously covered by ice sheet (which is represented by land in the model). These changes are now clearly shown in Figure 1c,d. We have rewritten and clarified this statement, using the new data.

17. I.158-159, ‘is driven by a southward expansion of CDW below the freshwater layer’ Why to exclude the impact of reduced AABW formation. Any modelling signal to indicate this?

The reduction in AABW formation is closely linked to the southward expansion of CDW. We do not suggest it is one feature or the other, rather both adjust to the addition of freshwater at the surface. We have added a sentence to clarify this.

18. In general, how deep is the ocean channel between the WAIS Island and Antarctic main land, when the ice is reduced? Then, how is the water exchange between the Weddell Sea and Amundsen Sea? This is also key to understand the full process.

This channel is relatively shallow, and the flow must cross depths of only a few hundred metres (see Figure 1). The total transport through the gateway is rather small. There is a **southward transport** of 0.7 Sv and 1.1 Sv in SL4.1 and SL7.1 respectively (Pacific to Atlantic sector), and a **northward transport** of 0.1 Sv and 1.0 Sv in COMB4.1 and COMB7.1 respectively (Atlantic to Pacific sector) through the gateway at 73°S, 86-80°W. These numbers have now been added to the manuscript.

19. Please add the information about the MOC anomalies between LIG and PI. Also important to understand the PI-to-LIG ocean change.

We have added an anomaly figure for the MOC change showing LIG minus PI to the Supplementary material (Figure S10), and briefly discuss the MOC changes between the LIG and PI in the manuscript.

Reviewer #3 (Remarks to the Author):

Review of NCOMMS-22-16471-T: “East Antarctic warming triggered by West Antarctic ice loss during the Last Interglacial”.

In this paper the authors discuss the impact of both WAIS removal and freshwater input on the climate and ocean state in the southern Hemisphere during the LIG. They use a climate model with a prescribed ice sheet and freshwater forcing. One issue they investigate is if the mismatch between the LIG127ka PMIP4 simulations and southern hemisphere proxy records (SAT and SST) was due to the choice of an PI AIS in the model setup. The authors focus on indefinited changes in southern hemisphere climate state specially from ice-sheet driven feedbacks (changes in bathymetry/elevation) from those related to freshwater by running three climate simulations.

The paper is well written, and the results are clearly presented. However, I would the authors to expand on previous publications which have investigate the combination of WAIS and FWF for the LIG, although they may be from older generation of climate models. For example, Holloway et al., 2016 also performed climate model simulations with a reduced AIS and FWF. (I was not associated with this study). I would recommend for publication with the minor changes below.

We thank the reviewer for the positive recommendation and constructive comments, which we have addressed below. As noted in our response to Reviewer 1, we now make comparisons to Holloway et al (2016).

Section 2: Question about lapse rate correction: I would like more details on the reasoning of this correction for the final COMB4.5 results (shown in the paper). The results in Fig3 are the SAT with the lapse-rate correction, so removing any warming signal associated with a change in altitude due to the retreat of the WAIS. However, as the aim of the SL4.5 is to investigate the impact of a reduced WAIS, this correction removes/reduces this impact. Why are the corrected results not shown in the SOM? I am not sure of the rationale? Is this to make the results more consistent

with FW4.5?

We reconsidered our approach, and we agree with the reviewer that it is better to present the uncorrected result in the manuscript. The lapse-rate correction is potentially interesting for analysing feedbacks. However, in light of our new experimental design and revised conclusions, the lapse-rate correction is no longer of central interest to this study and we have removed it.

Minor text corrections:

Line 35: “Coupling a dynamic ice sheet with ocean-atmosphere-sea-ice-land ... remains a significant technical barrier”. There have been simulations with dynamic ice sheet (Sommers et al, 2021). To be clearer, I would state a dynamic Antarctic ice sheet. This is a challenge due to the ice-ocean coupling required for a large marine based ice sheet.

We have adjusted the text to “dynamic Antarctic ice sheet”, and agree that ice-ocean coupling is a major challenge for climate models, which motivates our current approach. We have also referenced a recent modelling effort to achieve ice-ocean coupling (Kreuzer et al, 2021), and progress towards coupling with a fully coupled climate model (Smith et al, 2021).

Results section2: SL4.5 experiment compared to COMB4.5

From the SL4.5 (Fig2d) there is a pronounced warming signal ($> 2\text{C}$) at 100-500m between 140W and 180W. What is the origin of this? It does not occur when running the COMB4.5 experiment, unlike most other signals (Wind speed, SAT)? Is there a 2nd feedback so the FWF dominates the response in the ocean?

In general, the freshwater perturbation response tends to dominate the ocean response in the combined case (both surface and subsurface), while the ice loss perturbation tends to dominate the atmospheric response over Antarctica. However, there is also a non-linear effect whereby the deep ocean warms more strongly in the combined case than the freshwater only case (despite the ice loss experiment causing a cooling of the deep ocean). We now discuss this compound effect on the deep ocean more clearly in the manuscript.

Discussion:

Line 160 (related to FigS6)” Moreover, the East Antarctic subsurface anomaly is more positive than in FW4.5, which partly reflects the warming caused by the continental ice sheet reduction” It could be the colour scale but comparing Fig.2e and f most of EAIS appear to surrounding by ‘white’ region, so a very similar temperature as in FW4.5 (so colder), apart from close to the Wilkes Basin.

First, we accept that the differences were small in the original version. In the updated experiments, there is a discernible increase in temperature around East Antarctica when comparing COMB4.1 with FW4.1.

Also, in FigS6, with deeper SST anomalies the ice sheet only simulation (FigS6d), there is a strong cooling along the coast.

The subsurface warming is primarily driven by (a) surface freshening slowing down AABW formation, leading to (b) a flattening of isoneutral slopes as shown in the new Figures 4 and 5, which then causes (c) relatively warmer circumpolar deep water to be exported further south. Note the absolute temperatures in Figure 4a, which show relatively warmer water in the deep at 30-40 °S, which can then travel upwards and southwards along the flattened neutral surfaces.

Line 182: "there are also areas for East Antarctica where reverse-sloping bedrock..." Can you mention which of these three regions coincide with simulated warming/cooling anomalies. It is interesting the large warming anomaly near the Wilkes SG Basin, but the Amery is associated with a cooling.

There is significant localised warming (up to 5 °C) and associated ice loss at the Amery ice shelf and Fimbul ice shelf. A smaller warming and ice loss are found at the Shackleton ice shelf (~1-2 °C). A lesser anomaly (0.5-1.5 °C) is also found at the Wilkes Subglacial Basin, along with a modest reduction in ice, consistent with proxy constraints mentioned in response to Reviewer 1.

Figures:

Figure 2-3. Could you increase the latitude extent to be consistent with SOM plots. The lateral extent of the warmer anomalies (i.e Fig.2d) is chopped off.

We have adjusted the latitudinal extent so that all polar plots in the results now extend to a cut-off of 40°S.

Figure 3: what are the circles? Are there the ice core sites? Are the anomalies larger than 5C? if so can you edit the scale bar to indicate this?

The circles are proxy-based temperature anomalies. Yes, there are some model anomalies that exceed 5 °C, and we have adjusted the colorbar to have triangular endings to indicate this. Note that none of the proxies exceed 5 °C though.

Methods:

Table: Model setup: could the authors add a table in the SOM of model set up (forcings), names; ice sheets(small/large); size of FWF etc. Just as summary for the model setup and boundary conditions.

We have added a table summarising the boundary conditions to the supplementary material.

Table: Proxy data: Throughout the text the authors compare the SAT/SST from the simulations to proxy data and other values from PMIP4 simulations. Can these be summarised in a table. For example, the 4 proxy ice core records (line 80); SST temperature records (line 25-26). Can the authors compare to other proxy records,

for example those shown in the Otto-Bliesner et al., 2021, summary PMIP4
LIG127ka paper.

We have added a table showing comparisons to SAT and SST proxy records, as in
Otto-Bliesner et al (2021).

Reviewer #1 (Remarks to the Author):

The authors have addressed all my previous major comments and I congratulate them on an interesting study providing new insights into the missing pieces in understanding LIG Antarctic climatic changes. I still have some remaining minor comments which should be addressed before publication (see below) but these should be resolved easily.

General comment:

The fact that including a meltwater-feedback worsens the match with SST-proxies in the Southern Ocean is somewhat glossed over and merits a more extensive discussion (see specific comments below). As of now the abstract highlights the improved match with SAT records while ignoring the increased mismatch with SST-proxies. Both should be mentioned.

I missed this in the last round of reviews, but I think there are a couple of SST reconstructions for the LIG (e.g. in Figure 5 in Capron et al. 2014 I count at least 5 SST reconstructions which fall in your domain shown in fig 2 and 3, I'm sure there must be more recent studies as well). In Figure 3 you show ice core reconstruction but no SST proxies (neither in Fig. 2). If it would be possible to include the SST-comparison (at least in the figure) as well that would be fantastic. Having now read further I see that you provide a comparison in supp tables 1+2. I think adding the points in the figures as well would be nice if possible.

Below I list some minor edits and comments (please note that line numbering refers to the track-changes document)

L1 : was ca. 1 to 9 ...

L5 : both from topographic changes and

L6 : we find that changes in surface elevation and gradients ...

L10: the combination of a diminished ice sheet and enhanced ...

L11 : and induces further warming ... both of which could increase ice sheet retreat

L25 : suggested by Southern Hemisphere ...

L59 : ... climate model, assuming an idealised flattening or removal of WAIS. In this study the oceanic response was not assessed in detail.

L63: Here, we assess ...

L75: ... is found in the EAIS ...

L79 : both the Weddell and Ross Sea warming ...

L116/117: Here you could speculate as to how the katabatic winds would change in response to the topographic changes. Maybe there are studies which have looked at that?

L189-194: I think this paragraph is a bit positivistic. From supp table 1 and 2 I gather that including the meltwater feedback actually leads to a negative SST anomaly while the proxies suggest SST-warming during the LIG. The elevation feedback alleviates this somewhat but the net effect is still a LIG cooling. So, while improving the match with ice core proxies the combined feedbacks actually worsens the match with respect to SST? This merits a little more discussion of the involved effects (i.e. what's missing in the model) and caveats.

L156: 1-2 degree decrease in what?

L243 There should be one or two sentences pointing towards the fact that this increases the mismatch to SST-proxies in the region (see comment above) and suggestions as to why that may be and what could be done to remedy this (e.g. resolution, missing feedbacks, fully (incl. ice sheets) coupled models).

Figure 1. colorbar label please indicate change of what?? I assume it is surface elevation from the figure caption? Also the colorbar is very large compared to the plots.

Figure 2 and 3: add a proper label to the colorbars (e.g. SST anomaly and SAT anomaly). I suggest to cut the polar plots a little more to the south as you don't discuss temp changes in Patagonia. Cutting at 60 South would make enlarge the Antarctic portion of the model output considerably, makes it easier to read. If you decide to include the SST proxies as well ignore the

comment with regard to cutting at 60 south.

Figures 4 & 5. Colorbars don't have labels.

Reviewer #3 (Remarks to the Author):

Thank you for the extensive replies to my comments. I would recommend this paper for publication

We thank the reviewers for their positive response, and Reviewer 1 (Dr Sutter) for his additional comments which have helped to improve the manuscript. As before, we outline our response to the comments in blue.

Regards,
David Hutchinson

Reviewer 1

The authors have addressed all my previous major comments and I congratulate them on an interesting study providing new insights into the missing pieces in understanding LIG Antarctic climatic changes. I still have some remaining minor comments which should be addressed before publication (see below) but these should be resolved easily. We thank Dr Sutter for his positive feedback.

General comment:

The fact that including a meltwater-feedback worsens the match with SST-proxies in the Southern Ocean is somewhat glossed over and merits a more extensive discussion (see specific comments below). As of now the abstract highlights the improved match with SAT records while ignoring the increased mismatch with SST-proxies. Both should be mentioned. Due to word limits, we were not able to add this statement to the abstract, but we instead removed the “improved match” with SAT proxies for balance. We now discuss the comparisons to both SAT and SST data in the Results and Discussion section.

I missed this in the last round of reviews, but I think there are a couple of SST reconstructions for the LIG (e.g. in Figure 5 in Capron et al. 2014 I count at least 5 SST reconstructions which fall in your domain shown in fig 2 and 3, I'm sure there must be more recent studies as well). In Figure 3 you show ice core reconstruction but no SST proxies (neither in Fig. 2). If it would be possible to include the SST-comparison (at least in the figure) as well that would be fantastic. Having now read further I see that you provide a comparison in supp tables 1+2. I think adding the points in the figures as well would be nice if possible.

We have overlaid on Figure 2 the SST proxy anomalies from the compilations of Capron et al (2017) and Hoffman et al (2017), as used in the PMIP intercomparison of Otto-Bliesner et al (2021), and additional new proxy data from Chandler and Langebroek (2021). Since Figure 2 presents SST model data, we feel this is the appropriate place to show SST proxies. In Figure 3 (now re-ordered to Figure 5) we are only showing surface air temperature (SAT), therefore we only use the SAT proxies from Antarctica here.

We have also updated our Supplementary tables to include the new Chandler and Langebroek (2021) data, with model-based estimates provided for each location.

Below I list some minor edits and comments (please note that line numbering refers to the track-changes document)

L1 : was ca. 1 to 9 ...

We have inserted “approximately”.

L5 : both from topographic changes and
Done

L6 : we find that changes in surface elevation and gradients ...
We have changed this to “changes in surface elevation”

L10: the combination of a diminished ice sheet and enhanced ...
Done

L11 : and induces further warming ... both of which could increase ice sheet retreat
Done

L25 : suggested by Southern Hemisphere ...
Done

L59 : ... climate model, assuming an idealised flattening or removal of WAIS. In this study the oceanic response was not assessed in detail.
Done

L63: Here, we assess ...
Done

L75: ... is found in the EAIS ...
Done

L79 : both the Weddell and Ross Sea warming ...
Done

L116/117: Here you could speculate as to how the katabatic winds would change in response to the topographic changes. Maybe there are studies which have looked at that?
We considered this suggestion, but since we don't have a strong argument for katabatic wind changes we prefer not to speculate here.

L189-194: I think this paragraph is a bit positivistic. From supp table 1 and 2 I gather that including the meltwater feedback actually leads to a negative SST anomaly while the proxies suggest SST-warming during the LIG. The elevation feedback alleviates this somewhat but the net effect is still a LIG cooling. So, while improving the match with ice core proxies the combined feedbacks actually worsens the match with respect to SST? This merits a little more discussion of the involved effects (i.e. what's missing in the model) and caveats.

We have added the following discussion on this subject:

“These comparisons indicate that the simulated SST are generally too low compared to the SST proxy data. Most of the Southern Ocean SST proxies indicate warming of at least several degrees, although a few sites also indicate strong cooling (Figure 2), with the proxy anomaly data ranging from -6.8 deg C to 11.5 deg C (Supplementary Table 1). In general, the best agreement is found in the ice loss only experiments (SL4.1 and SL7.1), while a poorer agreement is found when introducing meltwater both separately (FW4.1 and FW7.1) and combined with the ice loss (COMB4.1 and COMB7.1). These data suggest that, in the context of uncertain timing of Antarctic melting at the LIG [Rohling et al. 2019; Barnett et al. 2023], that the peak meltwater pulse in the Southern Ocean did not occur simultaneously

with the warmest SST anomalies, since meltwater tends to strongly cool the surface of the Southern Ocean.”

L156: 1-2 degree decrease in what?

L156 did not include the words “1-2 degree decrease”, but L256-257 stated that “...anomalies are found ... (1-2 deg C)” without explicitly saying of which field. Therefore, I updated that statement to “SAT anomalies are found... (1-2 deg C)”.

L243 There should be one or two sentences pointing towards the fact that this increases the mismatch to SST-proxies in the region (see comment above) and suggestions as to why that may be and what could be done to remedy this (e.g. resolution, missing feedbacks, fully (incl. ice sheets) coupled models).

We have added the following:

“This cooling generally does not agree with Southern Ocean SST proxies from the LIG, which may indicate that the peak in meltwater from Antarctic ice melt at the LIG possibly succeeded the peak positive SST anomalies in the Southern Ocean. We note, however, that Southern Ocean SST proxies are almost all located between 40 and 50 deg S, while most of the SST anomalies simulated in our experiments are located poleward of 50 deg S, where no proxy records are available.”

Figure 1. colorbar label please indicate change of what?? I assume it is surface elevation from the figure caption? Also the colorbar is very large compared to the plots.

We have changed the colorbar titles in Figure 1 to “Change in elevation (m)” and “Change in depth (m)”. We have also slightly reduced the size of the colorbars.

Figure 2 and 3: add a proper label to the colorbars (e.g. SST anomaly and SAT anomaly). I suggest to cut the polar plots a little more to the south as you don't discuss temp changes in Patagonia. Cutting at 60 South would make enlarge the Antarctic portion of the model output considerably, makes it easier to read. If you decide to include the SST proxies as well ignore the comment with regard to cutting at 60 south.

In Figure 2 and 3 we updated the colorbar labels to “SST anomaly” and “SAT anomaly” respectively. In Figure 2, we decided to include the SST proxies as suggested by the reviewer, and therefore we retain the larger domain (up to 45 deg S), rather than cutting to 60 deg S. NOTE: Figure 3 (SAT anomaly) has now been re-ordered to Figure 5, to match the order of reference in the manuscript.

Figures 4 & 5. Colorbars don't have labels.

We have added colorbar titles to Figure 4 and 5.

NOTE: Figure 4 and 5 (MOC and zonal mean temperatures) have been re-ordered to Figure 3 and 4, to match the order of reference in the manuscript.